# Intrinsic signaling pathways modulate targeted protein degradation

Yuki Mori[1,2,8], Yoshino Akizuki[1,2,8], Rikuto Honda[1,2], Miyu Takao[2], Ayaka Tsuchimoto[1,2], Sota Hashimoto[3], Hiroaki Iio[3], Masakazu Kato[3,4], Ai Kaiho-Soma[1], Yasushi Saeki [5,6], Jun Hamazaki [3], Shigeo Murata [3], Toshikazu Ushijima [7], Naoko Hattori[7] & Fumiaki Ohtake [1,2] ✉

Targeted protein degradation is a groundbreaking modality in drug discovery; however, the regulatory mechanisms are still not fully understood. Here, we identify cellular signaling pathways that modulate the targeted degradation of the anticancer target BRD4 and related neosubstrates BRD2/3 and CDK9 induced by CRL2[VHL]- or CRL4[CRBN]-based PROTACs. The chemicals identified as degradation enhancers include inhibitors of cellular signaling pathways such as poly-ADP ribosylation (PARG inhibitor PDD00017273), unfolded protein response (PERK inhibitor GSK2606414), and protein stabilization (HSP90 inhibitor luminespib). Mechanistically, PARG inhibition promotes TRIP12-mediated K29/K48-linked branched ubiquitylation of BRD4 by facilitating chromatin dissociation of BRD4 and formation of the BRD4–PROTAC–CRL2[VHL] ternary complex; by contrast, HSP90 inhibition promotes BRD4 degradation after the ubiquitylation step. Consequently, these signal inhibitors sensitize cells to the PROTAC-induced apoptosis. These results suggest that various cell-intrinsic signaling pathways spontaneously counteract chemically induced target degradation at multiple steps, which could be liberated by specific inhibitors.

Targeted protein degradation, in which small-molecule degraders induce rapid degradation of disease-causing proteins, is an emerging concept in drug discovery[1]. The degraders, such as PROTAC hetero-bifunctional molecules or molecular glues, induce proximity of the target protein (neosubstrate) and the E3 ubiquitin ligase to cause forced ubiquitylation and subsequent degradation of the target through the proteasome[2]. The cullin-RING ubiquitin ligases (CRLs) CRL2[VHL] and CRL4[CRBN] are most prevalently used as E3s recruited by PROTACs[3]. Several PROTACs, such as the estrogen receptor degrader

ARV471 and the androgen receptor degrader ARV-110, have entered clinical trials for the treatment of breast and prostate cancers, respectively[4,5]. Given the further application of PROTACs for clinical use, a precise understanding of the regulatory mechanisms for targeted protein degradation is needed. Among the representative neosubstrates targeted by PROTACs, BET proteins, including BRD2, BRD3, and BRD4, are master regulators of transcription. BRD4 recognizes acetylated histones via its bromodomains and recruits p-TEFb and RNA polymerase II to facilitate active transcription[6]. Inhibitors of BET

[1]Laboratory of Protein Degradation, Institute for Advanced Life Sciences, Hoshi University, 2-4-41 Ebara, Shinagawa-ku, Tokyo 142-8501, Japan. [2]Graduate School of Pharmacy and Pharmaceutical Sciences, Hoshi University, 2-4-41 Ebara, Shinagawa-ku, Tokyo 142-8501, Japan. [3]Laboratory of Protein Metabolism, Graduate School of Pharmaceutical Sciences, The University of Tokyo, 7-3-1 Hongo, Bunkyo-ku, Tokyo 113-8656, Japan. [4]Faculty of Pharmaceutical Sciences, Teikyo Heisei University, Nakano-ku, Tokyo 1648530, Japan. [5]Division of Protein Metabolism, The Institute of Medical Science, The University of Tokyo, 4-6-1, Shirokanedai, Minato-ku, Tokyo 108-8639, Japan. [6]Protein Metabolism Project, Tokyo Metropolitan Institute of Medical Sciences, 2-1-6 Kamikitazawa, Setagaya-ku, Tokyo 156-8506, Japan. [7]Department of Epigenomics, Institute for Advanced Life Sciences, Hoshi University, 2-4-41 Ebara, Shinagawa-ku, Tokyo 142-8501, Japan. [8]These authors contributed equally: Yuki Mori, Yoshino Akizuki. ✉e-mail: f-ohtake@hoshi.ac.jp

proteins, including JQ1, are considered promising anticancer drug seeds, and BET protein degraders are more potent than inhibitors in cancer cell death and antiproliferation in both cell and mouse models[7].

The ubiquitylation of substrate proteins is achieved through a sequential reaction of ubiquitin-activating (E1), -conjugating (E2), and -ligating (E3) enzymes[8]. The ubiquitin chains can be linked through seven lysine residues, the first methionine, or certain serine/threonine residues[9]. The K48- or K11-linked ubiquitin chains are known as proteasomal targeting signals. In addition, ubiquitin chains of different linkage types can be branched (branched ubiquitin chains) to transduce unique signals. The K11/K48-, K48/K63-, or K29/K48-branched ubiquitin chains serve as strong proteasomal degradation signals[10]. Such diverse ubiquitin codes are written by linkage type–specific E2s and/or E3s and are decoded by specific proteins containing ubiquitin-binding domains. For degradation, ubiquitin chains are recognized by the proteasomal ubiquitin receptors, proteasome-associated deubiquitinases (DUBs), shuttling proteins such as Rad23, or the ubiquitin-dependent segregase/unfoldase p97/VCP (valosin-containing protein)[10]. Thus, the ubiquitin chain length, linkage types, and the structural properties of the substrates (e.g., the distribution of disordered regions) are determinants of substrate degradation[11].

The degradation of the neosubstrate induced by PROTACs proceeds through unique mechanisms that are in part common to but different from the degradation mechanism of genuine substrates. Formation of the stable substrate–PROTAC–E3 ternary complex is the key to efficient ubiquitylation[12]. Certain E2 enzymes, such as UBE2G and UBE2R, and neddylation cycle regulators help target ubiquitylation by CRLs[13–15]. After initiation of ubiquitylation, TRIP12 cooperates with CRL2[VHL] or CRL4[CRBN] to assemble K29/K48-linked branched ubiquitin chains and promote the degradation of certain neosubstrates[16]. p97/VCP can mediate targeted protein degradation through substrate segregation and unfolding[17]. Nevertheless, the regulatory mechanisms underlying targeted protein degradation are not fully elucidated. Especially, cellular signaling pathways affecting targeted protein degradation remain largely unexplored.

In this study, we hypothesize that cell-intrinsic pathways might enhance or repress these regulatory steps of targeted protein degradation. By combining the HiBiT-based screening of small molecules and biochemical mechanistic analyses, we identify certain inhibitors of cellular pathways that robustly promote PROTAC-induced targeted degradation of BRD family neosubstrates. These results will lead to a better understanding of the mechanisms of the targeted protein degradation.

## Results

### Various cellular pathways modulate the targeted degradation of BRD4

To identify the signaling pathways modulating targeted degradation, we screened chemicals that might modulate the degradation of a model neosubstrate, BRD4, in the presence or absence of the CRL2[VHL]-based PROTAC MZ1[18] (Fig. 1a). To this end, we established an HCT116 cell-derived cell line in which the HiBiT tag was knocked-in in the N-terminus of an endogenous BRD4 locus (Supplementary Fig. 1a). We confirmed that MZ1 effectively decreased HiBiT-BRD4 signals in a dose- (Fig. 1b) or time-dependent manner (Fig. 1c) and that the degradation was reversed either by the proteasomal inhibitor carfilzomib or by b-AP15, the inhibitor of the proteasome-associating DUBs UCH37 and USP14 (Supplementary Fig. 1b). We then conducted a screening of chemicals, where candidate chemicals were pre-treated 4 h before treatment with MZ1 (30 nM, 2 h), which itself induced subtle degradation of BRD4. By testing representative known signal inhibitors (Supplementary Data 1) using the HiBiT-BRD4 screening system, we identified candidate chemicals that enhance or inhibit the targeted degradation of BRD4 (Fig. 1d, e).

The chemicals that inhibit the degradation include the proteasome inhibitors carfilzomib and bortezomib (Fig. 1e, stained red), ensuring the rationality of our screening. Although a majority of the investigated chemicals altered the abundance of BRD4 irrespective of the presence or absence of MZ1, certain chemicals specifically enhanced the MZ1-induced degradation of BRD4 (Fig. 1e, stained purple). They include the PARG [poly-ADP ribosylation (PAR) glycosylase] inhibitor PDD00017273[19], the PERK inhibitor GSK2606414[20], and the HSP90 inhibitor luminespib[21] (Fig. 1e). We confirmed that these chemicals indeed enhance the degradation of HiBiT-BRD4 in a concentration-dependent manner in the presence of MZ1 but do not affect the BRD4 levels in the absence of MZ1 (Fig. 1f). These chemicals also promote BRD4 degradation induced by the CRL4[CRBN]-based degrader dBET6[22] (Fig. 1g). However, the abundance of BRD4 in the presence of the BRD4 inhibitor JQ1 is not affected by these chemicals, indicating that these chemicals specifically target the PROTAC-induced BRD4 degradation (Supplementary Fig. 1c). Collectively, these results introduce the possibility that various cell-intrinsic signaling pathways might spontaneously repress the targeted degradation of BRD4, which can be liberated by specific inhibitors.

### PARG inhibition facilitates targeted degradation of BRD4 and BRD2/3 but not MEK1/2 or ERα

We next analyzed the degradation of endogenous BRD4 in HeLa cells. Treatment with MZ1 at a suboptimal concentration and duration was used hereafter to evaluate the degradation-enhancing effect of compounds. We found that the PARG inhibitor PDD00017273 (hereafter, PDD) enhanced MZ1-induced degradation of BRD4, whereas it did not affect the steady-state levels of BRD4, CUL2, or VHL (Fig. 2a and Supplementary Fig. 1d). Cellular PAR levels were consistently upregulated by PDD (Fig. 2a); however, the concentration of PDD used (3 μM) is far lower than that reported to exhibit cell toxicity[19]. PDD enhanced BRD4 degradation at a concentration at which it does not affect cell viability (Fig. 2b). We also confirmed that a 6-h treatment with PDD did not significantly affect the expression levels of ubiquitin ligases or proteasome components, or the BRD4 levels, in RNA-Seq analysis (Fig. 2c and Supplementary Data 2). Knockdown of PARG also consistently promoted MZ1-induced degradation of BRD4 (Fig. 2d), and we confirmed the effect of PDD on BRD4 degradation in other cell lines such as HT1080 cells (Supplementary Fig. 1e).

PDD17273 enhanced BRD4 degradation induced by another CRL2[VHL]-based PROTAC, ARV771[23] (Fig. 2e). Interestingly, PDD also enhanced the BRD4 degradation induced by the CRL4[CRBN]-based PROTAC dBET6 (Fig. 2f), consistent with the HiBiT assay result (Fig. 1g). This result implies that the effect of PARG inhibition on BRD4 degradation is attributable to the neosubstrate (BRD4) rather than to the E3 (VHL/CRBN). We, therefore, investigated whether PARG inhibition regulates targeted degradation of other neosubstrates and found that degradation of ER-α induced by ARV471[5], as well as degradation of MEK1/2 by MS432[24], was not affected by PDD (Fig. 2g, h). These results indicate that PARG inhibition does not promote the efficacy of all the PROTACs but rather promotes the degradation of select neosubstrates.

BRD4 is known to be efficiently degraded by PROTACs such as MZ1, whereas degradation of BRD2 and BRD3 is relatively difficult[18]. Therefore, we reasoned that the effect of the degradation enhancer is more profound toward these hard-to-degrade targets. Indeed, we found that PDD17273 robustly enhanced MZ1- or ARV771-dependent degradation of BRD2 and BRD3 (Fig. 2a, e), which are only subtly degraded by the MZ1/ARV771 treatment alone (lanes 3,5, and 7).

Promotion of BRD4 degradation by PDD was inhibited by the proteasome inhibitor MG132, the ubiquitin E1 inhibitor TAK243, or the NEDD8 E1 inhibitor MLN4293 but not by the lysosomal inhibitor bafilomycin A (Fig. 2i). These results indicate that PDD promotes BRD4 degradation through the proteasome.

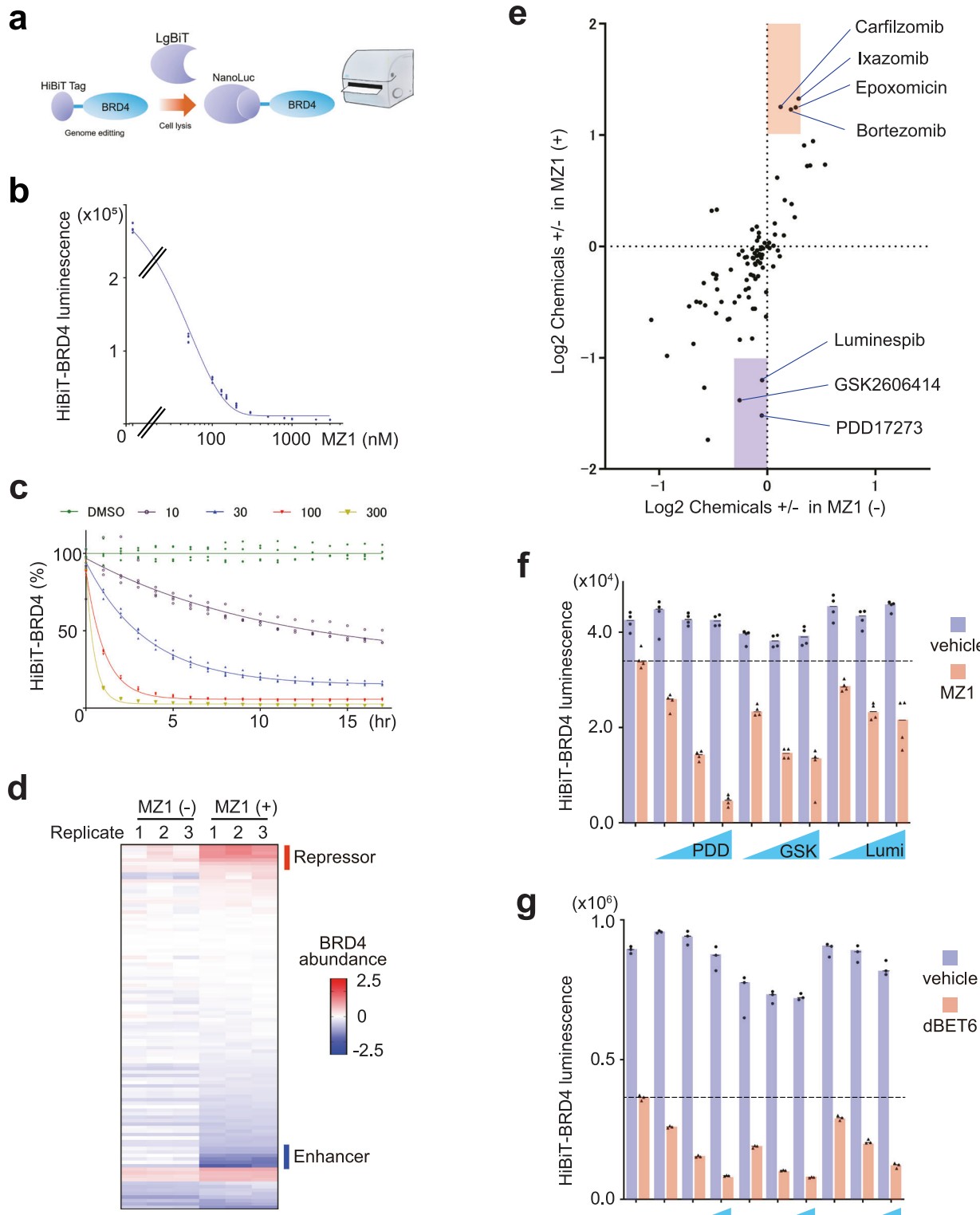

**Fig. 1 | Screening of pathways modulating targeted degradation of BRD4.**
**a** Scheme of screening using HiBiT-BRD4–expressing cells in (**d**, **e**). **b**, **c** MZ1-dependent decrease of HiBiT luminescence in a HCT116 cell line expressing HiBiT-BRD4. HiBiT-BRD4 cells were treated with the indicated concentration (nM) of MZ1 for 2 h (**b**) or for the indicated number of hours (**c**) before HiBiT luminescence analysis ($n = 5$ (**b**) or 4 (**c**), biological replicates). **d** Heatmap presentation of chemicals that enhance or repress BRD4 degradation ($n = 3$, biological replicates).

HiBiT-BRD4 cells were treated with candidate chemicals for 6 h and with 30 nM MZ1 for 2 h. **e** Chemicals that enhance HiBiT-BRD4 degradation in the presence of MZ1. The data in (**d**) are presented to analyze chemicals that enhance/repress the BRD4 degradation in the presence (*y*-axis) or absence (*x*-axis) of MZ1 ($n = 3$, biological replicates). **f**, **g** HiBiT-BRD4 cells were treated with PDD, GSK, or luminespib (1, 3, or 10 µM) for 4 h and with 30 nM MZ1 (**f**) or 50 nM dBET6 (**g**) for 2 h ($n = 4$ (**f**) or 3 (**g**), biological replicates).

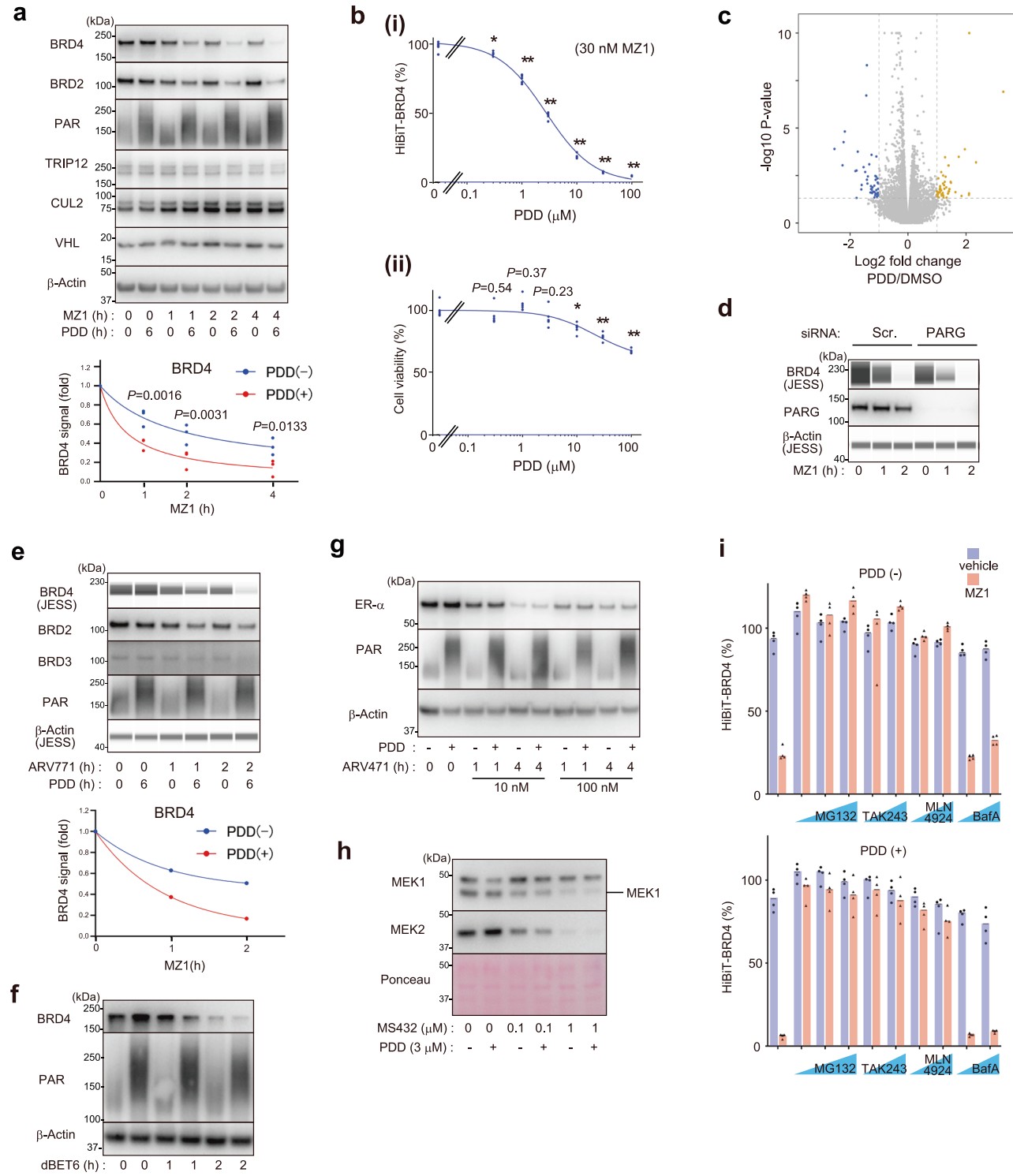

**Fig. 2 | PARG inhibition promotes targeted protein degradation of BRD4. a** PDD promotes MZ1-induced degradation of BRD4 and BRD2. HeLa cells were treated with 3 µM PDD and/or 100 nM MZ1 for the indicated number of hours. The lower panel shows the band intensities of the blots from three biological replicates. *P* values in ANOVA are shown. **b** PDD promotes BRD4 degradation at a lower concentration than the concentration at which it exhibits cytotoxicity. (i) HiBiT-BRD4 cells were treated with the indicated concentration of PDD (6 h) together with MZ1 (2 h), and BRD4 degradation was quantified (*n* = 6, biological replicates). Data were normalized to the MZ1 treatment alone. Asterisk: * *P* = 0.0003 or ** *P* < 0.0001 in ANOVA. (ii) The parental HCT116 cells were treated with the indicated concentration of PDD for 3 days, and cell viability was quantified (*n* = 5, biological replicates).

Asterisk: **P* = 0.0003 or ***P* < 0.0001 in ANOVA. **c** HCT116 cells were treated with 3 µM PDD for 6 h, and total RNA was isolated and subjected to RNA-sequencing analysis (*n* = 3, biological replicates). **d** Knockdown of PARG promotes BRD4 degradation. HT1080 cells were transfected with the indicated siRNAs for 3 days, then treated with MZ1 as indicated. **e**, **f** PDD promotes BRD4 degradation induced by different PROTACs. HeLa (**e**) or HCT116 (**f**) cells were treated with PDD and/or ARV771 (**e**) or with the CRL4^CRBN-based dBET6 (**f**), as indicated. **g**, **h** Targeted degradation of ERα or MEK1 is not affected by PDD. MCF7 (**g**) or HCT116 (**h**) cells were treated with the indicated chemicals. **i** HiBiT-BRD4 cells were pre-treated with either PDD (4 h) and/or the indicated inhibitors (0.5 h) and then with 30 nM MZ1 for an additional 2 h (*n* = 4, biological replicates).

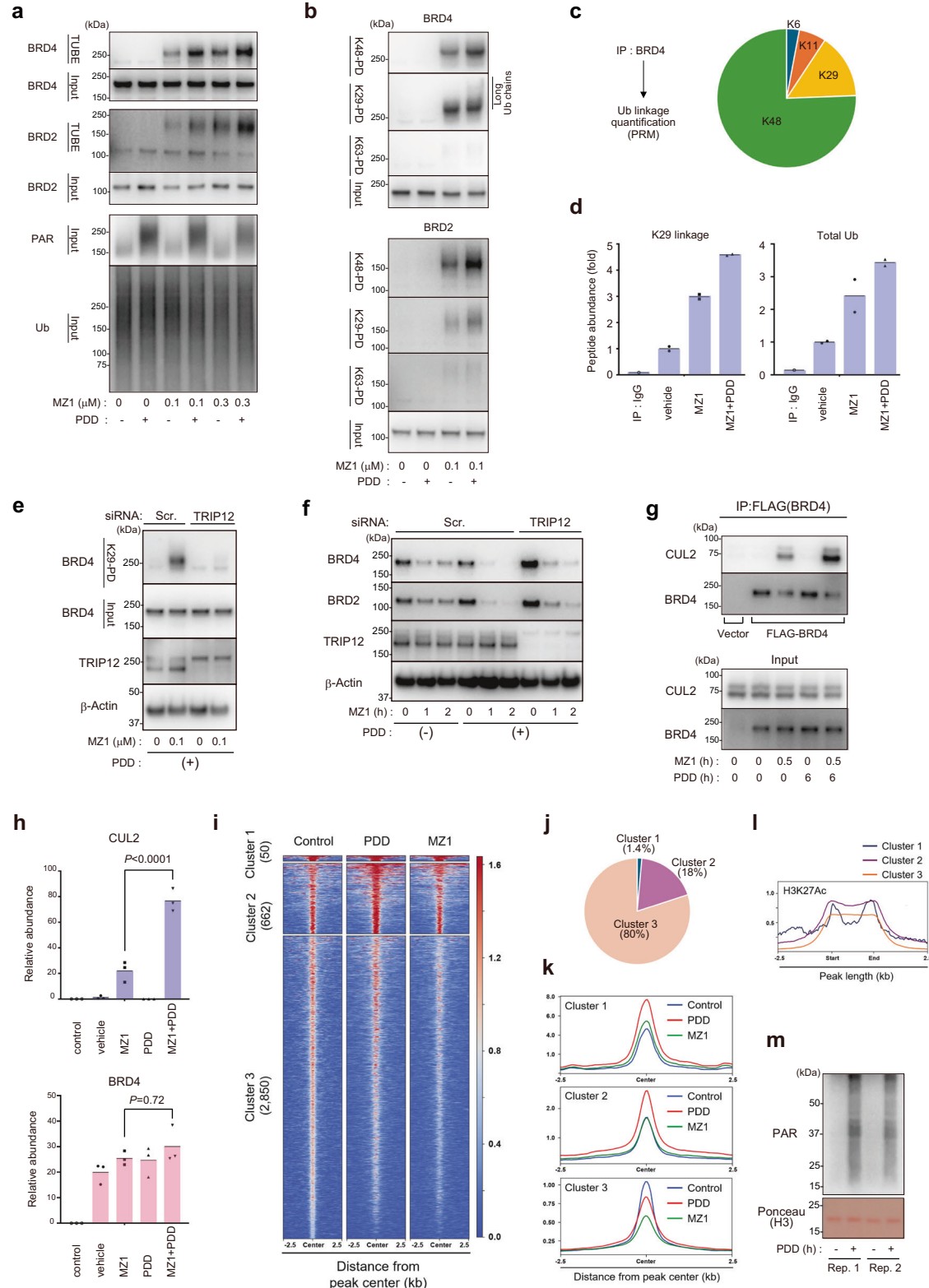

## PARG inhibition facilitates BRD4−MZ1−CRL2VHL ternary complex formation and K29/K48-branched ubiquitylation

Having shown that PARG inhibition by PDD robustly promotes PROTAC-induced degradation of BRD4, we next investigated whether the action of PDD is exerted before or after the ubiquitylation step. To this end, we purified ubiquitylated proteins from cells using a tandem ubiquitin-binding entity (TUBE2, consisting of the UBA domain of HR23A[25]). As shown in Fig. 3a, b, treatment with PDD enhanced the MZ1-dependent ubiquitylation of BRD4 and BRD2 (Fig. 3a). Notably, BRD4 is a high−molecular-weight protein with an apparent size of ~200 kDa; thus, ubiquitylated BRD4 species appeared as a stacked smear corresponding to ~250 kDa.

In a previous study, we showed that BRD4 is modified mainly with K29/K48-branched polyubiquitin chains[16], which is achieved through chain initiation and K48-linked ubiquitylation by CRL2VHL and the incorporation of K29-linked branched linkages by TRIP12. We

**Fig. 3 | PARG inhibition facilitates BRD4–MZ1–CRL2$^{VHL}$ ternary complex formation and K29/K48-branched ubiquitylation. a** PDD promotes ubiquitylation of BRD4 and BRD2. HeLa cells were treated with the indicated concentration of MZ1 for 1 h. MG132 was added 5 min prior to the treatment with MZ1. The ubiquitylated proteins were purified from the cell lysates using TUBE-conjugated agarose. Input or TUBE-pulldown samples were subjected to western blotting, as indicated. **b** TUBE pulldown using either K29-, K48-, or K63-specific TUBEs. **c** BRD4 was modified with K48- and K29-linked ubiquitin chains. Endogenous BRD4 was immunopurified from HCT116 cells treated with PDD (3 μM, 6 h) and MZ1 (100 nM, 1 h) together with MG132 (20 μM, 1 h) and subsequently subjected to PRM-based ubiquitin linkage quantification ($n = 2$, biological replicates). **d** HCT116 cells were treated with PDD (3 μM, 6 h) and/or MZ1 (100 nM, 1 h), and BRD4-modified ubiquitin chains were analyzed using PRM. The data show abundance (normalized to the vehicle) of signature peptides for K29 ubiquitin linkages and total ubiquitin ($n = 2$, biological replicates). **e** HT1080 cells transfected with the indicated siRNAs

were treated as in (**a**, **b**) to pulldown K29-linked ubiquitin chains. **f** Knockdown of TRIP12 partially canceled PDD-dependent promotion of BRD4 degradation. HeLa cells were transfected with the indicated siRNAs and treated with PDD (3 μM, 6 h) and/or MZ1 (100 nM, 1 or 2 h). **g** BRD4–MZ1–CRL2$^{VHL}$ ternary complex assembly. 293T cells were transfected with FLAG-BRD4 (lanes 2–5) and/or HA-VHL (lanes 1–5), and cell lysates were subjected to immunoprecipitation using anti-FLAG antibody. **h** The samples in (**g**) were subjected to LC-MS and label-free quantification ($n = 3$, biological replicates, ANOVA). **i** Heatmaps of ChIP-seq signals of BRD4 in the cells treated either with control, PDD17273, or MZ1. BRD4-binding regions in control cells ($n = 3562$) were subtracted for plotting, and peaks were divided into three clusters. **j** Percentage of BRD4-binding regions consisting of the three clusters. **k** Metaplots of ChIP-seq signals of BRD4 over the center of peaks. **l** Metaplots depicting H3K27ac ChIP-seq signals in HCT116 cells over BRD4-binding regions. **m** HCT116 cells were treated with PDD (5 μM, 4 h), and cell fractionation was performed. PARylation of chromatin fractions was analyzed.

therefore investigated whether endogenous BRD4 was modified with K29/K48-branched ubiquitin chains using K29-, K48-, or K63-linkage–specific binders (K29-binder, K48-TUBE, or K63-TUBE, respectively). Note that the K29-binder consists of the ubiquitin-binding domain (NZF1) of Trabid and is also capable of purifying K33-linked chains[26]; however, we confirmed that BRD4-modified ubiquitin chains merely contain this linkage type[16]. We found that MZ1-induced modification of BRD2 and BRD4 with K48- and K29-linked chains was enhanced by PDD (Fig. 3b). For the K29 linkages conjugated to BRD4, PDD promotes the abundance of longer ubiquitin chains (marked in Fig. 3b). Mass spectrometric quantification of ubiquitin linkages[27,28] revealed that immunopurified endogenous BRD4 in the presence of MZ1 + PDD is mainly modified with K48-linked (70%) and K29-linked (20%) ubiquitylation (Fig. 3c). Moreover, we found that treatment with PDD enhanced the K29-linked ubiquitin chains modifying BRD4 (Fig. 3d). Because TRIP12 inserts K29 linkages into CRL2$^{VHL}$-mediated K48-linked chains to assemble K29/K48-branched ubiquitin chains[16], we investigated whether TRIP12 is involved in this process. We found that K29-linked ubiquitylation of BRD4 in the presence of MZ1 + PDD was abolished by the TRIP12 knockdown (Fig. 3e). Moreover, the knockdown of TRIP12 partially reversed the targeted degradation of BRD4 enhanced by the co-treatment with MZ1 + PDD (Fig. 3f).

Previous literature showed that the coprecipitation of CUL2 with BRD4 serves as an effective indicator of the BRD4–PROTAC–CRL2$^{VHL}$ ternary complex formation in cell[29]. Therefore, we carried out a co-immunoprecipitation analysis with anti-FLAG (BRD4) antibody and found that the MZ1-induced interaction of CUL2 with BRD4 was further enhanced by PDD (Fig. 3g and Supplementary Fig. 2a). We also conducted a label-free quantification of the FLAG-BRD4 immunocomplex using LC-MS. The association of CUL2 with BRD4 was substantially enhanced by the co-addition of PDD (Fig. 3h). We analyzed the intracellular localization of BRD4 as a possible mechanism for the increased ternary complex formation. Because the JQ1 moiety in MZ1 and other BRD4-targeting PROTACs mimic acetylated lysine to target bromodomains (BD1 and/or BD2) of BRD4, which associate with the acetylated histone[6], we reasoned that chromatin association of BRD4 might compete with the ternary complex formation. We therefore conducted cell fractionation, in which a soluble nuclear fraction is extracted by high-salt treatment, and the proteins tightly associating to the chromatin are subsequently released after micrococcal nuclease treatment. We found that BRD4 tightly associated with chromatin (Supplementary Fig. 2b, lanes 5 and 11) is decreased after treatment with PDD (lanes 6 and 12).

To investigate whether excessive PARylation decreases the chromatin association of BRD4, we conducted ChIP-Seq analyses[30] using anti-BRD4 antibody. The cells were treated with either control or PDD17273, or with MZ1 as a positive control, and soluble chromatin was subjected to immunoprecipitation. First, the BRD4-binding regions in control cells were divided into three clusters (Fig. 3i, j). In

clusters 1 and 2, which accounted for 1.4 and 18% of the total binding regions, respectively (Fig. 3j), MZ1 did not induce chromatin dissociation of BRD4 (Fig. 3k). These regions were highly enriched with histone H3K27 acetylation (Fig. 3l). The data suggested that clusters 1 and 2 are "MZ1-hyporesponsive" regions, in which BRD4 appears to tightly associate with high levels of H3K27Ac. However, in cluster 3, which accounted for 80% of the total binding regions (Fig. 3j), MZ1 induced eviction of BRD4 from chromatin (Fig. 3k, l). These regions were "MZ1-responsive" regions.

Strikingly, in cluster 3 (MZ1-responsive regions), PDD17273 massively decreased chromatin association of BRD4 (Fig. 3k). The data supported our model that excessive PARylation decreases chromatin association of BRD4. Because cluster 3 is ~80% of the total BRD4-binding regions, we assume that the PDD-induced chromatin eviction of BRD4 in cluster 3 is the main contributor to the enhanced BRD4 ubiquitylation and degradation.

We also found that, in clusters 1 and 2 (MZ1-hyporesponsive regions), PDD rather increased chromatin association of BRD4 (Fig. 3k). Although the reason for this behavior is currently unknown, excessive PARylation might alter chromatin structures to allow BRD4 accessibility at certain chromatin domains. These BRD4 associating with the cluster 1/2 regions are not likely to contribute to the MZ1-induced degradation, at least within a short timecourse.

We also confirmed that PDD induces hyper-PARylation of chromatin proteins (Fig. 3m). Collectively, our data suggest that the PARG inhibitor PDD promotes targeted degradation of BRD2/3/4 via enhanced BRD–PROTAC–CRL2$^{VHL}$ ternary complex formation and K29/K48-linked ubiquitylation. As an underlying mechanism, we propose that PARG inhibition induces dissociation of BRD4 from chromatin, which enhances the accessibility of BRD4 to form the ternary complex, leading to increased ubiquitylation (Supplementary Fig. 2c).

## Proteostatic pathways regulate BRD4 degradation through multiple steps

From the screening in Fig. 1, we noticed that inhibitors of proteostatic pathways—the PERK inhibitor GSK2606414 (hereafter, GSK) and the HSP90 inhibitor luminespib—were identified as the BRD4 degradation enhancers (Fig. 1d, e). PERK is one branch of the unfolded protein responses (UPRs), whose inhibition causes global accumulation of misfolded proteins[31]. Similarly, HSP90 is one of the master regulators of cellular protein homeostasis; it functions by associating with and refolding aberrantly misfolded proteins[32]. We confirmed that several PERK inhibitors (GSK, GSK2656157, and AMG-PERK) or HSP90 inhibitors (luminespib and 17-AAG) enhance MZ1-induced degradation of endogenous BRD4 or BRD2 (Fig. 4a, b). The effect of GSK on BRD4 degradation is mediated through the proteasome (Fig. 4c). Although HSP90 inhibition results in the destabilization of certain proteins such as Ras[32], the steady-state protein level of BRD4 in the absence of MZ1

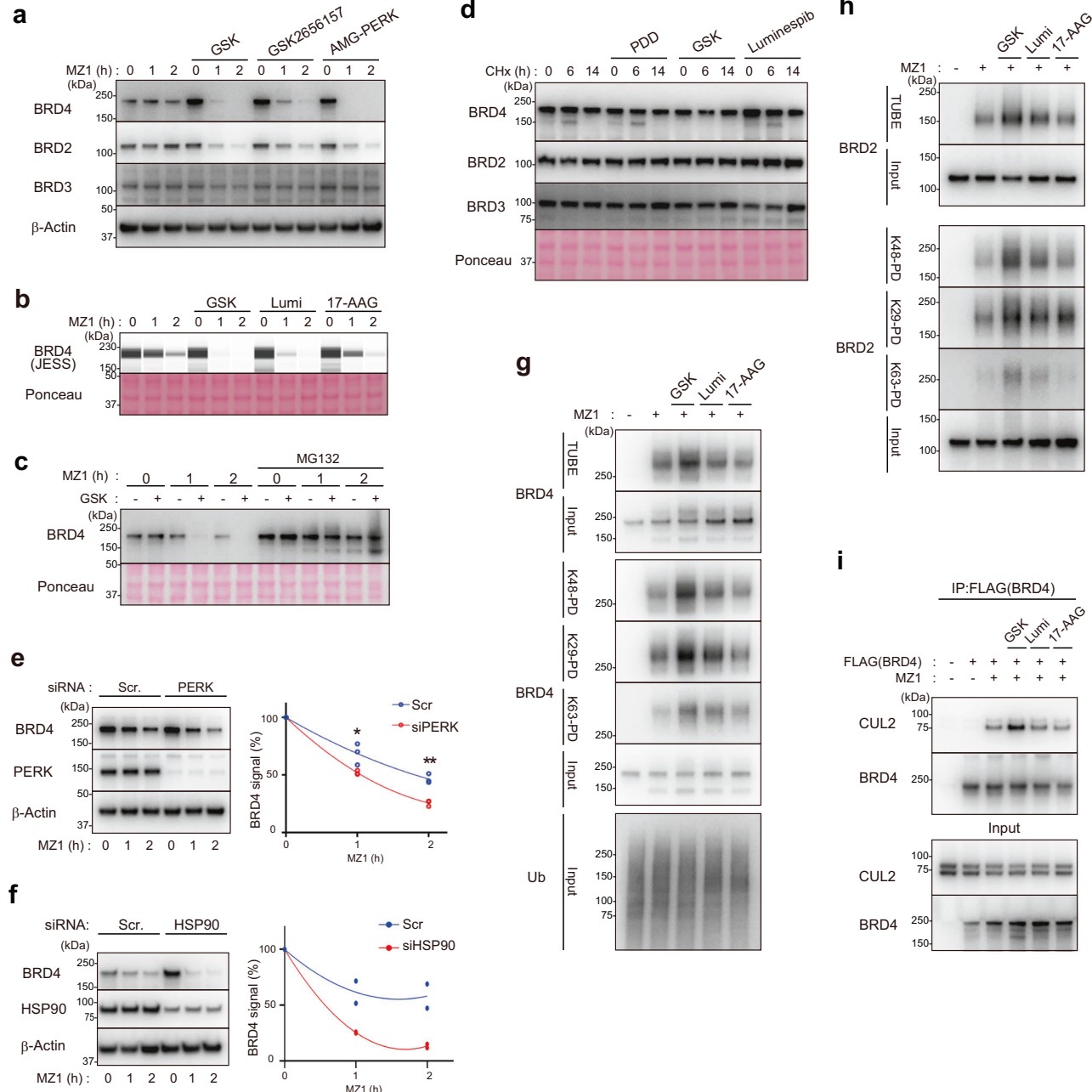

**Fig. 4 | Proteostatic pathways promote BRD4 degradation through multiple steps. a** PERK inhibitors promote BRD4 degradation. HCT116 cells were treated with either GSK, GSK2656157, or AMG-PERK (10 µM, 6 h) and/or MZ1 (100 nM, 1 or 2 h). **b** HSP90 inhibitors promote BRD4 degradation. HCT116 cells were treated with either GSK, luminespib, or 17-AAG (10 µM, 6 h) and/or MZ1 (100 nM, 1 or 2 h). **c** BRD4 degradation in the presence of GSK is proteasome-dependent. HCT116 cells were treated with either GSK (10 µM, 6 h), MG132 (20 µM, 2 h), or MZ1 (100 nM, 1 or 2 h). **d** HCT116 cells were treated with either PDD, GSK, or luminespib (10 µM, 14 h) and/or 50 ng/mL CHX for the indicated number of hours.

Total cell lysates were subjected to Western blotting. **e, f** HCT116 cells were transfected with the indicated siRNAs and treated with MZ1 (100 nM, 1 or 2 h). (Right) BRD4 band intensities were quantified (*n* = 3 (**d**) or 2 (**e**), biological replicates). Asterisk: **P* = 0.0014 or ***P* = 0.0002 in ANOVA. **g, h** HCT116 cells were treated with either GSK, luminespib, or 17-AAG (10 µM, 6 h) and/or MZ1 (100 nM, 1 h). Cell lysates were subjected to pulldown using the indicated TUBEs. **i** 293T cells were transfected with FLAG-BRD4 (lanes 2–6) and/or HA-VHL (lanes 1–6) and then treated with the indicated chemicals; co-immunoprecipitation was subsequently performed.

was not decreased by either PERK or HSP90 inhibition (Fig. 4a, b). We analyzed a basal turnover of BRD2/3/4 by cycloheximide (CHX) chasing. We found that both BRD4 and BRD2/3 are highly stable in the steady state (Fig. 4d). After CHX incubation for 6 and 14 h, treatment of cells with PDD, GSK, or luminespib did not change the half-life of BRD2/3/4 (Fig. 4d). These data suggest that, within the technically available time ranges, the investigated signal inhibitors do not influence the basal half-life of BRD2/3/4. We also confirmed that

knockdown of either PERK or HSP90 consistently accelerated the MZ1-induced degradation of endogenous BRD4 (Fig. 4e, f).

We next explored the mechanism by which the proteostatic pathways enhance targeted degradation. To investigate whether these inhibitors enhance ubiquitylation of BRD2 and BRD4, we purified ubiquitylated proteins using binders for pan-ubiquitin (TUBE2) or specific ubiquitin linkages (K29-, K48-, or K63-linkage–specific binders). Our previous study established that the majority of ubiquitin

linkages conjugated on BRD4 upon MZ1 treatment are K29- and K48-linked chains[16]. We found that MZ1-induced total ubiquitylation, as well as K29- or K48-linked ubiquitylation, of BRD2 and BRD4 was enhanced by GSK (Fig. 4g, h). By contrast, although luminespib and 17-AAG exerted a level of enhancement in BRD4 degradation comparable to that of GSK, HSP90 inhibition by luminespib or 17-AAG minimally affected BRD4/2 ubiquitylation (Fig. 4g, h). Moreover, the PERK inhibitor GSK enhances BRD4−MZ1−CRL2[VHL] ternary complex formation, whereas HSP90 inhibitors minimally affect the ternary complex assembly (Fig. 4i). These results indicate that HSP90 inhibition affects a step downstream of BRD4/2 ubiquitylation.

To gain insight into the mechanism of action of HSP90 inhibition, we conducted a co-immunoprecipitation analysis from the 293T cells expressing FLAG-RAD23B. Neither the interaction of RAD23B with the proteasome nor the interaction of RAD23B with the ubiquitylated proteins was changed by HSP90 inhibition (Supplementary Fig. 2d). These results suggest that HSP90 inhibition does not globally affect the shuttling protein RAD23B delivery of ubiquitylated proteins to the proteasome.

Collectively, these results indicate that inhibitors for PERK or HSP90, both of which are major proteostatic pathways, modulate chemically induced degradation of BRD4 by affecting different steps of the ubiquitin-proteasome pathway.

### PDD17273 and GSK2606414 facilitate PROTAC-induced apoptotic cell death

We analyzed the cellular consequence of enhanced degradation of BRD4/2/3. Because MZ1-induced BRD4 depletion is known to cause cancer cell death[18], we quantified the caspase 3/7 activity (Fig. 5a) and the cell viability (Fig. 5b). We found that HeLa or HCT116 cancer cell lines are sensitized to MZ1-induced apoptosis (Fig. 5a) and cell death (Fig. 5b) by the co-treatment with MZ1 and either PDD or GSK, whereas treatment with PDD alone does not affect cell viability at the investigated concentration (Fig. 5a, b). Notably, GSK treatment alone led to a small decrease in cell viability; we therefore analyzed the percentages by which the cell number decreased after treatment with MZ1. As shown in supplementary Fig. 3a, GSK co-treatment substantially sensitizes cells to MZ1 in the MZ1 concentration range of 30−300 nM. We also confirmed PDD/GSK-dependent modulation of apoptotic responses by immunostaining apoptotic markers (Fig. 5c, d). Notably, a substantial portion of BRD2 escaped degradation even after a 24-h treatment using MZ1 alone, whereas co-treatment with either PDD or GSK resulted in degradation of most of the BRD2 (Fig. 5c, d). We therefore propose that simultaneous degradation of BRD2/3/4 contributes to the effect of PDD/GSK treatment on cell death. We note that HSP90 inhibitors are reported to exhibit cytotoxicity even in the absence of MZ1[21], which hampers further characterization of the enhancement effects. We also note that HCT116 cells were previously used for the analysis of cell death induced by BRD4-targeting PROTACs[33]. We confirmed that melanoma-derived mouse B16 cells are also sensitized to MZ1- or ARV771-induced cell death by the co-treatment with either PDD or GSK (Supplementary Fig. 3b−e).

To further investigate the effect of PDD or GSK on the MZ1-induced apoptosis, we performed RNA-Seq analysis. A multidimensional scaling analysis revealed that treatment with PDD alone does not induce a large alteration in global gene expression but enhances the gene expression alteration induced by MZ1 (Fig. 5e). By contrast, treatment with GSK alters a subset of genes irrespective of the presence or absence of MZ1 (Fig. 5e, y-axis) yet also enhances the effect of MZ1 (Fig. 5e, x-axis). A hierarchical clustering analysis reveals that MZ1-up/downregulated genes are further up/downregulated by either PDD or GSK (Fig. 5f). Indeed, MZ1-induced repression of proliferation markers MYC and HER2 (ERBB2), as well as induction of growth arrest/p53 activation markers p21 and OSGIN1, was further enhanced by either PDD or GSK (Fig. 5g). JQ1-induced cell death was

not enhanced by the co-treatment of PDD/GSK (Supplementary Fig. 3f, g). Moreover, neither the JQ1-induced increase of p21 and OSGIN expression, nor the decrease of ERBB2 and MYC, was further enhanced by the co-treatment of PDD/GSK (Fig. 5h). The list of upregulated/downregulated genes induced by GSK or PDD analyzed by RNA-Seq is provided in the Supplementary Data 3.

### PDD17273 and GSK2606414 enhance the degradation induced by a high-performance PROTAC SIM1

We investigated whether the PROTAC enhancer chemicals can promote targeted degradation induced by highly potent degraders. SIM1 is a recently developed trivalent PROTAC that more potently induces the formation of the BRD−PROTAC−CRL2[VHL] ternary complex and subsequent degradation of BRD4 as well as BRD2/3[34]. SIM1 induced the rapid degradation of BRD4 with a DC$_{50}$ of ~4.2 nM after 2 h treatment (Fig. 6a, b). We found that the co-treatment with either PDD17273 or GSK further enhanced SIM1-induced degradation with a half-maximal degradation concentration (DC$_{50}$) of ~0.57 nM (Fig. 6a−c). Consistently, co-treatment using SIM1 + PDD or SIM1 + GSK induced cell death more efficiently than the treatment using SIM1 alone, with a ~10-fold lower IC$_{50}$ (Fig. 6d, e). These results suggest that PARG or PERK signaling pathways modulate targeted degradation of BRD family proteins induced by different types of PROTACs.

### Signal inhibitors regulate the targeted degradation of other neosubstrates

Finally, to investigate whether these signal inhibitors also promote certain other neosubstrates, such as kinases, we first focused on CDK9, a chromatin-localized kinase regulating RNAPII-mediated transcription[35]. We found that PDD17273 enhanced CDK9 degradation induced by a CRBN-based PROTAC ThalSNS (Fig. 6f). Moreover, PDD17273 significantly enhanced the ThalSNS-induced cell death in the ThalSNS concentration range 30−500 nM (Fig. 6g). These results suggest that PARG inhibition is effective for not only degrading BRD family proteins but also degrading another chromatin regulator, CDK9. We found that GSK co-treatment also enhanced the targeted degradation of CDK9 (Fig. 6h).

A previous study reported that HSP90 inhibition destabilizes the kinase c-MET and that the co-treatment of HSP90 inhibitor with PROTAC shows an additive effect on c-MET degradation[36]. As previously reported for c-MET and ER-α[36], we found that luminespib decreased steady-state levels of c-MET, ER-α, and CDK9 (Fig. 6h−j). The co-treatment of luminespib and PROTACs exhibited an additive effect on their degradation (Fig. 6h−j). Collectively, these results suggest that enhancement of targeted degradation by signal inhibitors can be applied to neosubstrates other than BRD family proteins.

## Discussion

In the present study, we screened a chemical inhibitor library for representative cellular signaling pathways and identified several pathways that can affect the targeted protein degradation (Fig. 6k). The prevailing view of targeted protein degradation is that a properly designed PROTAC induces the formation of the neosubstrate−PROTAC−E3 ternary complex and that the neosubstrate is readily ubiquitylated and degraded through harnessing the conventional proteasomal pathway[2]. Our results provide an additional view that various cell-intrinsic pathways spontaneously counteract target degradation at multiple steps. This leads to a better understanding of the mechanism for targeted protein degradation and may lead to the identification of yet other signaling pathways regulating the targeted degradation. Indeed, we have shown that inhibitors to PARG, PERK, or HSP90 robustly enhance the targeted degradation of BRD4 and sensitize cells to PROTAC-induced apoptosis. Moreover, enhancement of targeted protein degradation can be applied to other substrates such as difficult-to-

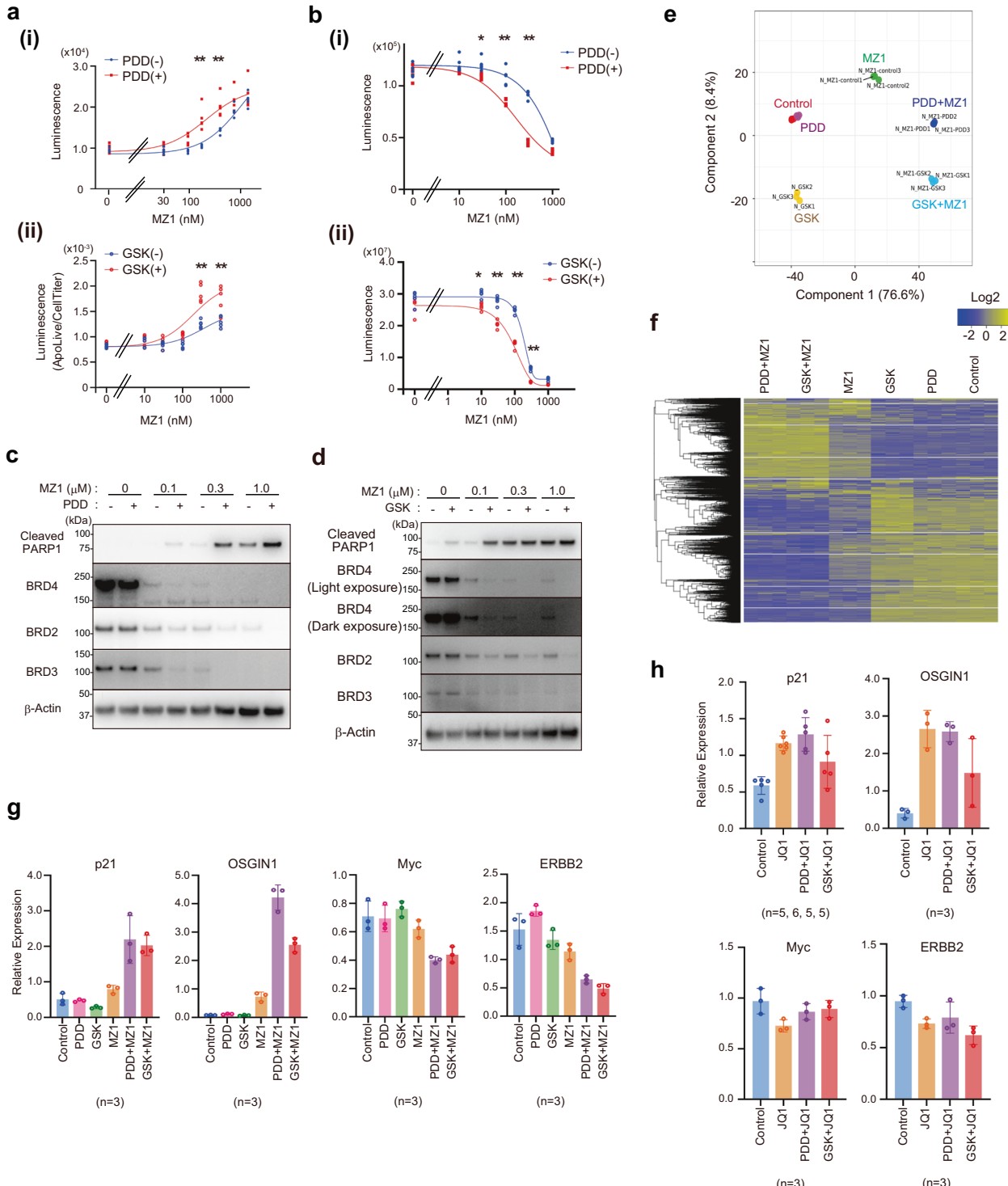

**Fig. 5 | Consequence of enhanced BRD4 degradation by inhibiting signaling pathways. a** HeLa (i) or HCT116 (ii) cells were treated with PDD or GSK together with the indicated concentration of MZ1 for 24 h, and caspase 3/7 activity was measured. Asterisk: **$P < 0.0001$ in ANOVA ($n = 5$, biological replicates). **b** HeLa (i) or HCT116 (ii) cells were treated with PDD or GSK together with the indicated concentration of MZ1 for 3 days, and cell viability was measured. Asterisk: (i) *$P = 0.0023$ or **$P < 0.0001$ in ANOVA ($n = 5$, biological replicates). (ii) *$P = 0.0001$ or **$P < 0.0001$ in ANOVA ($n = 5$, biological replicates). **c, d** HeLa (**c**) or HCT116 (**d**) cells were treated as in (**a**), and cell lysates were subjected to western blotting. **e, f** RNA-sequencing analysis. HCT116 cells were treated with either PDD, GSK, or MZ1 for 24 h, and total RNA was isolated. Multiple dimension analysis (**e**) and clustering analysis (**f**) are shown ($n = 3$, biological replicates). **g, h** HCT116 cells were treated with either vehicle, PDD, GSK, or MZ1 (**g**) or JQ1 (**h**) for 24 h, and total RNA was isolated. Quantitative RT-PCR was performed ($n$ is indicated in each panel; biological replicates). Error bars show SD.

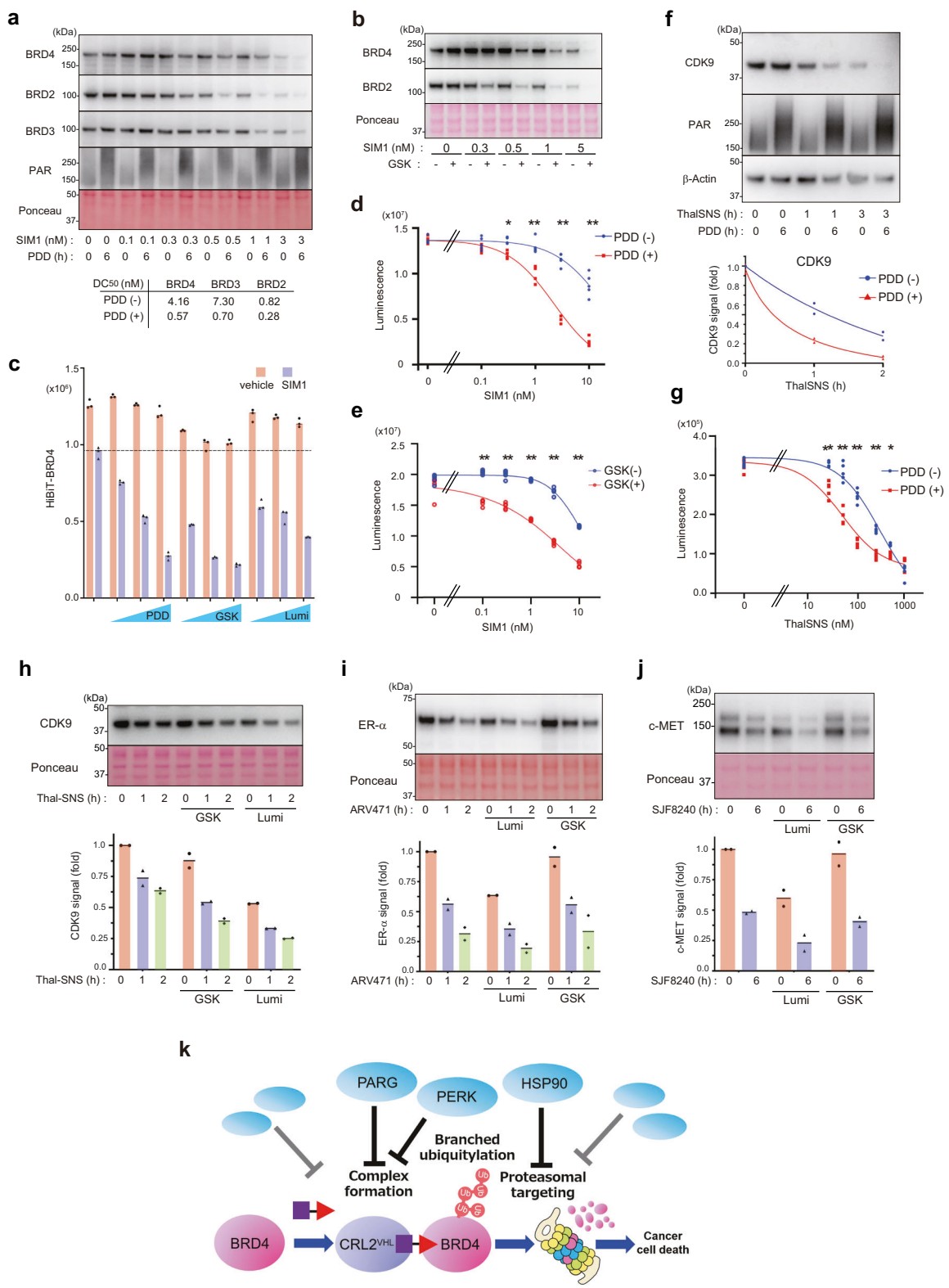

degrade neosubstrates BRD2 and BRD3, another chromatin regulator CDK9, and several HSP90 clients such as c-MET and ER-α. In this regard, inhibitors of the drug transporter MDR1 are recently reported to enhance the efficacy of the PROTAC-induced degradation of MEK and FAK among other neosubstrates[37]. By contrast, PDD did not promote the degradation of MEK, suggesting that the signal inhibitors identified in the present study act in a manner specific to the neosubstrate. Our results imply that signaling pathways affecting

targeted protein degradation are diverse, depending on the neosubstrates. Because we have established a framework for identifying such signaling pathways, the strategy presented here might lead to the identification of new pathways using other neosubstrates in general.

We further analyzed the molecular mechanism by which signal inhibitors regulate targeted protein degradation by focusing on the PARG inhibitor PDD. Enhanced BRD4 degradation due to PARG

**Fig. 6 | Application of enhancing targeted protein degradation. a**, **b** PDD or GSK promotes BRD4 degradation induced by SIM1. HeLa (**a**) or HCT116 (**b**) cells were treated with PDD, GSK, and/or MZ1 as indicated, and cell lysates were subjected to western blotting. **c** HiBiT-BRD4 cells were treated as indicated, and luminescence was measured (n = 3, biological replicates). **d**, **e** PDD or GSK promotes cell death induced by SIM1. HeLa cells were treated with PDD, GSK, and/or MZ1 for 3 days, as indicated. Asterisk: (**d**) *P = 0.0006 or **P < 0.0001 in ANOVA (n = 5, biological replicates). **e** **P < 0.0001 in ANOVA (n = 5, biological replicates). **f**, **h**–**j** HeLa (**f**), HCT116 (**h**), MCF7 (**i**), or MDA-MB231 (**j**) cells were treated with the indicated chemicals for 6 h or for the indicated number of hours (100 nM ThalSNS, 30 nM ARV471, 0.5 μM SJF8240). Total cell lysates were subjected to Western blotting. The

lower panel shows the band intensities of the blots from two biological replicates. **g** HeLa cells were treated with the indicated chemicals for 3 days, and cell viability was quantified. Asterisk: *P = 0.026 or **P < 0.0001 in ANOVA (n = 5, biological replicates). **k** Schematic model. Various cell-intrinsic pathways spontaneously counteract target degradation at multiple steps. Inhibitors to PARG, PERK, or HSP90 robustly enhance the targeted degradation of BRD4 as well as BRD2/3 and sensitize cells to PROTAC-induced apoptosis. PARG inhibition promotes TRIP12-mediated K29/K48-branched ubiquitylation of BRD4 by facilitating the BRD4-PROTAC-CRL2^VHL ternary complex, while HSP90 inhibition promotes BRD4 degradation after the ubiquitylation step.

inhibition is attributed to BRD4 ubiquitylation. Our data suggest that K29/K48-branched ubiquitylation and the cognate E3 ligase TRIP12 are involved in the enhanced degradation of BRD4 in the presence of MZ1 + PDD. Branched ubiquitin chains serve as a priority signal for the degradation of proteasomal substrates, such as misfolded or hard-to-degrade substrates[10], and have recently emerged as a ubiquitin code that enhances chemically induced targeted protein degradation[16]. Collectively, these data suggest that the K29/K48-branched ubiquitin chains play an important role in promoting the chemically induced degradation of unnatural, otherwise stable neosubstrates. With respect to the underlying mechanism, our data suggest that PARG inhibition promotes neosubstrate–PROTAC–E3 ternary complex formation without affecting the expression levels of CUL2 or VHL. PARG inhibition facilitates ternary complex formation by weakening the BRD4 association with chromatin. Using ChIP-Seq analysis, we showed that excessive chromatin PARylation due to PARG inhibition indeed accelerates BRD4 dissociation from chromatin. This accelerated dissociation likely facilitates the binding of PROTACs to BRD4, leading to enhanced ternary complex formation. We note that the results of the cell fractionation experiment (Supplementary Fig. 2b) and the ChIP-Seq experiment differ, presumably because (1) these methods examine global vs. region-specific changes of BRD4 binding and (2) cell fractionation uses a high-salt solubilization buffer to extract soluble nuclear extract, thereby enabling the extraction of weakly associating chromatin proteins.

We subsequently analyzed the action of the PERK inhibitor GSK and the HSP90 inhibitors luminespib and 17-AAG to show that certain compounds other than PDD also act as enhancers of targeted protein degradation, thereby demonstrating the usability of our proposed strategy. The detailed mechanisms of action of these compounds are future research topics to be rigorously investigated. We presume that PERK might regulate the expression of certain factors involved in the chromatin association of BRD4. Interestingly, HSP90 inhibitors promote BRD4 degradation but do not affect its polyubiquitylation efficiency, indicating that HSP90 inhibitors promote targeted degradation after the ubiquitylation step. Because HSP90 associates with BRD4 at the steady state (Supplementary Fig. 3h), HSP90 might stabilize BRD4 after it is ubiquitylated. Importantly, multiple pathways regulate proteasomal protein degradation. For example, PKA phosphorylates proteasomal subunits[38] and mTOR inhibition activates proteasomal activity[39]. Upstream regulators of DUBs that might counteract targeted protein degradation should also exist. Therefore, we anticipate that our subsequent screening might identify numerous cellular pathways, whose identification would be beneficial for elucidating the mechanism of targeted protein degradation.

Finally, although more work is needed, our data may lead to a new concept in degrader drug discovery that inhibiting these cell-intrinsic inhibitory pathways may improve the efficacy of targeted protein degradation. Alternatively, the expression levels of certain components of these signaling pathways might be useful to predict the efficacy of targeted degradation therapies. Moreover, pan-enhancer chemicals, which might promote targeted protein degradation in general, are an interesting topic for future study.

## Methods

### Plasmids and siRNAs
Mammalian expression vectors for human BRD4, CUL2, and VHL have been described previously[16]; site-directed mutagenesis was performed using PrimeStar Max (Takara). Ambion Silencer Select siRNAs for TRIP12 (s17809), PARG (s16158), PERK (s18102), HSP90 (s6995, s7001), and a scrambled siRNA (Silencer Select Negative Control #1: 4390843) were purchased from Thermo Fisher Scientific.

### Antibodies
The following antibodies were used for western blotting: anti-BRD4 (Cell Signaling Technology, #13440), anti-BRD2 (Cell Signaling Technology, #5848), anti-BRD3 (Santa Cruz Biotechnology, sc-515666), anti-PAR (Cell Signaling Technology, #83732), anti-TRIP12 (Proteintech, #25303-1-AP), anti-CUL2 (Santa Cruz Biotechnology, sc-166506), anti-VHL (Novus Biologicals, #091504), anti-b-Actin (Santa Cruz Biotechnology, sc-47778), anti-PARG (Cell Signaling Technology, #66564), anti-ER-α (HC-20; Santa Cruz Biotechnology, sc-543), anti-MEK1 (Cell Signaling Technology, #9124), anti-MEK2 (Cell Signaling Technology, #9125), anti-Ub (P4D1; Santa Cruz Biotechnology, sc-8017), anti-HSP90 (Santa Cruz Biotechnology, sc-69703), anti-ETS1 (Cell Signaling Technology, #14069), anti-PERK (Cell Signaling Technology, #3192), anti-CDK9 (Cell Signaling Technology, #2316), anti-c-MET (Cell Signaling Technology, #4560), Rad23B (Cell Signaling Technology, #13525), PSMB5 (Cell Signaling Technology, #12919), PSMD4 (Cell Signaling Technology, #3846), and anti-Cleaved PARP1 (Cell Signaling Technology, #5625). For immunoprecipitation, anti-BRD4 (Abcam ab128874), or anti-FLAG (Sigma-Aldrich, M2, #A2220) antibody was used.

### Chemicals
MZ1 (LifeSensors, PC1001), ARV771 (MedChemExpress, HY-100972), dBET6 (MedChemExpress, HY-112588), JQ1 (Selleck S7110), PDD17273 (Tocris, 5952), GSK2606414 (Selleck, S7307), GSK2656157 (Selleck, S7033), AMG-PERK (Selleck, S2853), luminespib (Selleck, S1069), 17-AAG (Sigma-Aldrich, A8476), ARV471 (MedChemExpress, HY-138642), SIM1 (R&D Systems 7432), MG132 (Santa Cruz Biotechnology), carfilzomib (Selleck, S2853), epoxomicin (Selleck, S7038), bortezomib (Selleck, S1013), iodoacetamide (Sigma-Aldrich), bafilomycin A (Sigma-Aldrich), b-AP15 (Sigma-Aldrich), TAK243 (Selleck, S8341), MLN4924 (Selleck, S7109), ThalSNS (Tocris, 6532), SJF8240 (Tocris, 7266), and protease inhibitor cocktails (Nacalai Tesque, 03969-34) were purchased from the indicated manufacturers.

### Cell lines, culture, and treatment
Human HCT116, HeLa, HT1080, MCF7, and HEK293T cells were obtained from ATCC and maintained at 37 °C with 5% $CO_2$ in Dulbecco's modified Eagle's medium (DMEM) high-glucose medium (Sigma-Aldrich) supplemented with 10% fetal bovine serum (FBS) (Sigma-Aldrich), penicillin–streptomycin (100 units/mL, GIBCO #15140148), sodium pyruvate (1 mM, GIBCO #11360070), and MEM nonessential amino acids (1×, GIBCO #11140050).

To generate HCT116-HiBiT-BRD4 knock-in cell lines in which the HiBiT tag was inserted into the N-terminus of BRD4, a CRISPR guide

sequence targeting exon 2 of BRD4 was cloned into pSpCas9(BB)−2A-Puro (PX459) v.2.0, which was a gift from F. Zhang (Addgene, #62988)[1]. The target sequence was 5′-ACTAGCATGTCTGCGGAGAG-3′. Puromycin-resistant clones were isolated and validated using western blot analysis and DNA sequencing.

## Cell lysis, immunoprecipitation, enrichment of ubiquitin chains, and immunoblotting

For the preparation of total cell lysates, cells were lysed in a lysis buffer [10 mM Tris-HCl (pH 7.5), 150 mM NaCl, 0.5 mM EDTA, 1% NP-40, 10% glycerol, and 0.5% Triton-X100] and extensively sonicated (Handy Sonic, TOMY Seiko). After centrifugation, protein concentrations were determined using a BCA kit (Takara). For co-immunoprecipitation, cells were lysed in a lysis buffer [10 mM Tris-HCl (pH 7.5), 150 mM NaCl, 0.5 mM EDTA, 1% NP-40, and 10% glycerol] supplemented with a protease inhibitor cocktail (Nacalai Tesque). Soluble cell lysates were incubated with antibodies at 4 °C for 4 h for immunoprecipitation. For immunoprecipitation under denaturing conditions, cells were lysed in a urea-containing lysis buffer [10 mM Tris-HCl (pH 7.5), 150 mM NaCl, 0.5 mM EDTA, 1% NP-40, 10% glycerol, and 6 M urea] supplemented with 10 μM MG132, 10 μM PR619, 10 mM iodoacetamide, and a protease inhibitor cocktail (Nacalai Tesque) and were then extensively sonicated (Handy Sonic, TOMY Seiko). The soluble lysates were diluted tenfold in lysis buffer before immunoprecipitation. For immunoblot assays, chemiluminescence signals were detected using FUSION (Vilber-Lourmat). For quantitative immunoblotting, the lysates were assayed using a JESS instrument (ProteinSimple).

Enrichment of total ubiquitin chains was performed using TUBE2 agarose (LifeSensors, #UM402); enrichment of K29-, K48-, or K63-linked ubiquitin chains and their conjugated proteins was performed using GST-Trabid-NZF1 (UBPBio, J4470), FLAG K48-TUBE HF (LifeSensors, #UM607), or FLAG K63-TUBE (LifeSensors, #UM604), respectively. For K48- or K63-TUBE pulldowns, cells were lysed in 150 μL of lysis buffer [100 mM Tris-HCl (pH 8.0), 150 mM NaCl, 5 mM EDTA, 1% NP-40, 0.5% Triton-X100, 10% glycerol] supplemented with 20 μM MG132, 10 μM PR619, 10 mM iodoacetamide, and a protease inhibitor cocktail (Nacalai Tesque) and were then extensively sonicated (Handy Sonic, TOMY Seiko). The lysates were diluted tenfold with reaction buffer [100 mM Tris-HCl (pH, 8.0), 150 mM NaCl, 5 mM EDTA, 10% glycerol]. One microgram of FLAG-tagged K48-TUBE (LifeSensors, #UM607) or FLAG-tagged K63-TUBE (LifeSensors, #UM604) was added to cell lysates to capture K48- or K63-linked chains, respectively, and was precipitated using anti-FLAG (Sigma-Aldrich, M2) affinity resin. For GST-Trabid-NZF1 pulldown, 20 μg of GST-Trabid-NZF1 (UBPBio, J4470) was added to cell lysates and was precipitated using glutathione sepharose (UBPBio, J4470).

## RT-quantitative PCR and RNA-sequencing

Total RNA was extracted using an RNeasy Mini Kit (QIAGEN, 74104). Reverse transcription reactions were performed using 400 ng of total RNA and a PrimeScript RT Master Mix (Takara, RR036A). The cDNA samples were diluted fivefold. RT-PCR assays were performed using a StepOnePlus Real-Time PCR system (Applied Biosystems, 4376592) with a TB Green Fast qPCR Mix (Takara, RR430) and gene-specific primers. Sequences were as follows: TBP-f: 5′-GAGCCAAGAGTGAAGAA-CAGTC-3′, and TBP-r: 5′-GCTCCCCACCATATTCTGAATCT-3′. p21-f: 5′-TCACTGTCTTGTACCCTTGTGC-3′, and p21-r: 5′-GGCGTTTGGAGTGG-TAGAAA-3′. OSGIN1-f: 5′-AACCCCATTGACGTGGACC-3′, and OSGIN1-r: 5′-CAAACCTCACGAAGTTGTCCC-3′. MYC-f: 5′-CCTGGTGCTCCATGAG-GAGAC-3′, and MYC-r: 5′-CAGACTCTGACCTTTTGCCAGG-3′. ERBB2-f: 5′-TGGCCTGTGCCCACTATAAG-3′, and ERBB2-r: 5′-AGGAGAGGTCAG-GTTTCACAC-3′. RNA sequencing was performed by Macrogen using Nova-Seq 6000 (Illumina). A library was prepared using the TruSeq stranded mRNA LT Sample Prep Kit (Illumina). For the volcano plot, the raw $P$ values, which were two-sided and not adjusted, were used.

## Quantification of cell viability, apoptosis, or HiBiT signal

Cell viability quantification was performed using the CellTiter-Glo Assay (Promega)[27]. HeLa or HCT116 cells were seeded on a 96-well plate at 5000 cells/well. The next day, the cells were treated with a vehicle or the indicated concentrations of compounds for 3 days. After cell lysis, the luminescent signal was quantified using the GloMax Discover System (Promega). Caspase 3/7 activity was measured using the ApoLive-Glo multiplex assay (Promega, G6410).

For the HiBiT analysis, HCT116-HiBiT-BRD4 cells were seeded on a 96-well plate at 10,000 cells/well. After 2 days, the cells were first treated with the indicated compounds for 4 h and then with 30 or 100 nM MZ1 for an additional 2 h. The HiBiT signal was measured using a Nano-Glo HiBiT Lytic Detection System (Promega, N3030). For detection of the HiBiT signal from live cells, a Nano-Glo Endurazine Live Cell Substrate (Promega, N2570) was used. For screening of compounds, HCT116-HiBiT-BRD4 cells were treated with 10 μM of compounds from a chemical library (Selleck) for 4 h and then treated with 30 nM MZ1 for an additional 2 h.

## Absolute quantification of ubiquitin linkages (Ub-AQUA/PRM)

Mass spectrometry-based quantification of the ubiquitin chains was performed as previously described[28,40] with minor modifications. After in-gel trypsin digestion, a mixture of AQUA peptides (25 fmol/injection) was added to the extracted peptides, and the concentrated peptides were diluted with 20 μL of 0.1% TFA containing 0.05% $H_2O_2$. For liquid chromatography–tandem mass spectrometry (LC-MS/MS) analysis, an Easy nLC 1200 (Thermo Fisher Scientific) was connected inline to an Orbitrap Fusion LUMOS (Thermo Fisher Scientific) equipped with a nanoelectrospray ion source (Thermo Fisher Scientific). Peptides were separated on a C18 analytical column (IonOpticks, Aurora Series Emitter Column, AUR2-25075C18A 25 cm × 75 μm, 1.6 μm FSC C18 with a nanoZero fitting) using a 50-min gradient (solvent A, 0.1% FA; and solvent B, 80% ACN/0.1% FA)[27]. For the targeted acquisition of MS/MS spectra (parallel reaction monitoring) for the ubiquitin chain–derived signature peptides, the Orbitrap Fusion LUMOS instrument was operated in targeted MS/MS mode using the Xcalibur software. The peptides were fragmented using higher-energy collisional dissociation (HCD) with a normalized collision energy of 28, and fragment ions were detected using an Orbitrap. Data were processed using the PinPoint software (Thermo Fisher Scientific), and peptide abundance was calculated on the basis of the integrated area under the curve (AUC) for the selected fragment ions.

## Chromatin immunoprecipitation sequencing (ChIP-seq)

HCT116 cells were treated either with vehicle, PDD (10 μM, 4 h), or MZ1 (30 nM, 1 h). Chromatin was cross-linked with 1% formaldehyde at room temperature for 10 min and sonicated in SDS-lysis buffer (1% SDS, 50 mM Tris-HCl pH 8.0, 10 mM EDTA pH 8.0, with protease inhibitor) using a Bioruptor (Cosmo Bio, Tokyo, Japan). After centrifugation, the supernatant was collected as a crude chromatin solution[30]. About 30 μg of sonicated chromatin was incubated with antibodies against BRD4 (Cell Signaling Technology, #13440, clone E2A7X, rabbit monoclonal, 10 μL) at 4 °C overnight. Chromatin was collected using Dynabeads Protein G (Thermo Fisher Scientific, Waltham, MA) and then eluted with TE buffer. After reverse cross-link in the eluate using 200 mM NaCl and proteinase K (Thermo Fisher Scientific), DNA was isolated by phenol and chloroform extraction and ethanol precipitation. For ChIP-seq analysis, the sequencing library was prepared with GenNext NGS Library Prep Kit (Toyobo) from 5 ng and sequenced using Illumina NextSeq2000 (Illumina, San Diego, CA) for a final sequencing depth of 50 million reads per sample. ChIP-seq data were submitted to the GEO database under accession #GSE262938. The Spike-in normalization strategy was performed to normalize all ChIP-seq data. About 20 ng of Spike-in chromatin (53083, Active Motif,

Carlsbad, CA) and 2 μg of Spike-in antibody (61686, Active Motif), were used according to the manufacturer's instructions.

### ChIP-seq data analysis

For sequencing QC and adapter trimming of all ChIP-seq data, fastp (version 0.23.4) was used. All reads were aligned to the human genome GRCh38/hg38 and the spike-in Drosophila genome dm6 using the short-read aligner Bowtie 2 (version 2.5.1) with default parameters. The resultant SAM file was converted into a BAM file, and duplicates were removed using SAMtools (version 1.18). Uniquely aligning Drosophila sequence tags were counted and compared to the sample containing the least number of tags to generate a normalization factor for random downsampling. The regions enriched with BRD4 were determined as narrow peaks using MACS2 (version 2.2.9.1) with a $p$ value threshold of $10^{-4}$. The ChIP-seq data for H3K27ac in HCT116 cells was downloaded (https://www.ncbi.nlm.nih.gov/; GSM2571028).

### Statistics and data reproducibility

Statistical analysis was performed using GraphPad Prism 7. The two-sided unpaired Student's $t$-test or the two-sided ANOVA test for biological replicates was used as indicated in the appropriate figure legends. Data show the means from biological replicates. The sample size ($n$) for each experiment is indicated in the corresponding legends. For Figs. 2d–h, 3a, b, e–g, m, 4a–c, g–i, 6a, b, the gel blot images are representatives of at least two independent experiments.

### Reporting summary

Further information on research design is available in the Nature Portfolio Reporting Summary linked to this article.

## Data availability

The raw datasets for RNA-sequencing analyses have been deposited to the GEO with the accession number GSE243615. The raw datasets for ChIP-sequencing analyses have been deposited to the GEO, with accession number GSE262938. The remaining data are available within the Article, Supplementary Information, or Source data file. Source data are provided with this paper.

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

## Acknowledgements

We thank Akinori Endo for preliminary experiments, Kentaro Shiraha and Ayami Hiratsuka for technical assistance, and Mikihiko Naito, Takuya Tomita, and Keiji Tanaka for discussions. BRD4 ChIP-seq was performed using NextSeq2000 (Illumina) under the support of the Early Life Stage of AMED-CREST/PRIME. This work was supported in part by JSPS KAKENHI (grant numbers JP21H02433, JP18H05498, and JP20K21408 to F.O.), AMED-CREST (grant number 21458950 to F.O.), the Takeda Science Foundation (to F.O.), and the Naito Foundation (to F.O.).

## Author contributions

F.O. designed the project and analyzed the data. Y.M. and Y.A. performed most of the cell-based and in vitro experiments and mass spectrometric analyses. R.H. and N.H. performed ChIP analysis. M.T., A.T., and A.K.-S. assisted with cell-based experiments. S.H., H.I., M.K., and J.H. provided reagents and preliminary experimental results. Y.S., S.M., and T.U. provided reagents and advice. F.O. wrote the paper. All authors discussed the results and commented on the manuscript.

## Competing interests

The authors declare no competing interests.
