## [Peer Review File · Nature Communications]

Reviewers' Comments:

Reviewer #1:

Remarks to the Author:

Mori et al. performed a chemical screen for enhancers or abrogators of BRD4 degradation using a HibiT Knock-In reporter cell line and the PROTAC MZ-1. They identified the PARG inhibitor PDD17273, luminespib and GSK2606414 as enhancers of BRD4 degradation and proteasome inhibitors as abrogators of BRD4 degradation and confirmed these with follow up studies. They demonstrate that PDD17273 is a BRD specific enhancer, via inhibition of PARG that functions via enhanced ubiquitination of BRD4, especially by TRIP12 induced K29/K48 branched ubiquitination, resulting from enhanced availability of BRD4. Additionally, GSK2606414 enhances degradation via PERK inhibition, resulting in enhanced ubiquitination and luminespib acts by HSP90 inhibition, post ubiquitination event. The authors go on to demonstrate that both PDD17273 and GSK2606414 co-treatment with MZ-1 results in enhanced apoptosis, reduced proliferation and enhanced anti-tumor effect in vivo. Finally, the authors demonstrate that these compounds also enhance the activity of more active BRD PROTAC SIM1.

This is an interesting and well controlled study into the effects of intrinsic cell signaling pathways and how they can be modulated to enhance the action of PROTACS. However, there are several issues which prevent it from being suitable for the general audience of Nature Communications in this form.

The impact of these discoveries is limited due to the fact they are limited to BRD family proteins. The authors must demonstrate that the additive effects of these compounds is general across other protein targets for TPD (Eg. Kinases) otherwise the interest is limited. As the authors say, PDD17273 is likely specific to BRD family proteins but PERK and HSP90 inhibitors are likely to be generally useful. Especially since it was reported several years ago, and is well known, that HSP90 inhibitors enhance PROTAC activity (PMID: 29129716) further reducing the novelty of this study.

Some of the conclusions drawn are not fully supported by data and should be further explored:

1. The authors state that PDD17273 increases ternary complex formation and measure this by Cul2 Co-IP – this is hard to conclude from the figure presented in Fig. 3G. The authors should explore this further, potentially by NanoBRET ternary complex formation assays (PMID: 34432243)
2. Further, the conclusions that increased PARylation evicts BRD4 from chromatin is insufficiently supported, additional experimentation (imaging, CHIP-Seq) is required to support this claim.

No protocol or method is presented for how the screen was performed and which compounds were screened – this must be included in the manuscript or SI for it to be useful to the scientific community. Compounds which failed to induce changes in BRD4 degradation could be as informative in understanding mechanisms, as those that did (depending on library composition).

4 points are shown on Fig. 1E for compounds which inhibited PROTAC activity but only 3 are named and all are proteasome inhibitors, what is the fourth compound? Depending on the compound library, several other compounds should be included in this group (eg. JQ1, MLN4924 etc.)

Statistics should be included in Fig. 5H to report the significance of this relatively subtle difference.

Reviewer #2:

Remarks to the Author:

Key results

The key message of this study centers around the enhancement of BRD4 (and BRD2/3)

degradation by VHL and CRBN PROTACs through targeted inhibition of the PAR and unfolded protein response pathways (PERK and HSPs). The authors performed a screen enabled by a stable HiBiT-Tag BRD4 cell line of various inhibitors. This is either a direct effect on BRDs through induction of their chromatin dissociation or by influencing degradation post-ubiquitination. These types of cell intrinsic effects may broadly apply to targeted degradation of specific substrates in general and allow for better mechanistic understanding in the field.

Validity

The story focused on the degradation of BRD4 is done appropriately, with the proper UPS controls and would be easily reproducible by readers of the study. The methodology they present can be applied more broadly to other neo-substrates. The data presented in the paper is of high quality and easy to follow, with significant impacts on protein degradation clear from the author's figures. How generalizable these conclusions are outside of Bromodomain proteins is unclear. While the steady state of the BRD proteins stays the same, changes to their intrinsic half-life by the GSK and PDD molecules may also have impacts on their degradation.

Significance

The concepts presented in this paper will have a significant impact on the scientific community as it seeks to elucidate the impacts of variable cell signaling events on targeted protein degradation. This touches on a critical biological aspect that influences the activity of PROTAC molecules distinct from previous studies that have looked at E3 ligase expression, target expression, and localization. Though the scope of the paper is narrowly focused on BRD4, the techniques can be easily applied to other targets. Understanding these pathways may impact our knowledge of the development of resistance to degraders and their treatment.

Data and methodology

Analytical approach

Your assessment of the strength of the analytical approach, including the validity and comprehensiveness of any statistical tests. If any aspect of the analytical approach is outside the scope of your expertise, please note this in your report or in the comments to the editor.

Suggested improvements

I think the authors should consider changing the title of this paper due to the bromodomain focus of the conclusions. If it included analysis of other PROTAC molecules from literature this would be more appropriate.

The authors demonstrate that the inhibitors do not affect the steady state levels of the bromodomain proteins, another consideration would be to investigate how the inhibitors influence the basal half-life of BRD2/3/4 with a cycloheximide chase or similar assay.

The authors show the synergy observed with co-treatment of GSK/PDD with MZ1 (Figs. 5b and S2). It would also be important to show this with JQ1 and GSK/PDD co-treatment as a control, as this should not result in the same synergistic effect if the synergy is due to degradation impacts vs downstream signaling. These controls would also be nice to see in the impact on downstream signaling in figure 5g.

More detail of the diversity of the screening library may be useful.

The mechanism of the enhancement of the VHL-BRD ternary complex in Fig. 4h is unclear, do you see the same linkage changes?

Correlating the enhancement of degradation by GSK or PDD with the expression of their respective targets may give insight to how the variance of these factors impact degradation.

Although outside of the scope of this work, analysis of genomic CRISPR screening data may provide further insights as to the most important pathways that enhance targeted protein degradation as this will cover more than the inhibitor library.

Clarity and context

The text of this paper is presented clearly.

References

This manuscript references previous literature appropriately.

Reviewer #3:

Remarks to the Author:

In this study, the authors identify cellular signaling pathways that modulate the targeted degradation of the anticancer target BRD4 and related hard-to-degrade targets BRD2/3 induced by CRL2VHL- or CRL4CRBN -based PROTACs. Several inhibitors of cellular signaling pathways were screened as degradation enhancers, such as poly-ADP ribosylation (PARG inhibitor PDD00017273), unfolded protein response (PERK inhibitor GSK2606414), and protein stabilization (HSP90 inhibitor luminespib). Among them, PARG inhibition promotes TRIP12-mediated K29/K48-linked branched ubiquitylation of BRD4 by facilitating chromatin dissociation of BRD4 and formation of the BRD4-PROTAC-CRL2VHL ternary complex; HSP90 inhibition promotes BRD4 degradation after the ubiquitylation step. In addition, these signal inhibitors sensitize cells to PROTAC-induced apoptosis. However, some issues should be addressed:

1. The proteasome inhibitor in Figure S1b is Carfilzomib, while the proteasome inhibitor mentioned in the paper is MG132.
2. No statistical analyses were made in Figure 2a and Figure 2b
3. In Figure 3a, the concentration of MZ1 changes, while the time of PDD changes. Multiple variables lead to unreasonable experimental design.
4. There was no statistical analysis of the changes of BRD4-MZ1-CRL2VHL ternary complex assembly in Figure 3g.
5. Please replace Figure 4h with a better-quality version where all values can be read clearly.
6. Excessive tumor volume at the end of treatment in a mouse allograft model.
7. The starting point of HCT116 cell viability assay in Figure 5b (ii) was significantly different when MZ1 concentration was 0 nM.

8. Please verify whether HSP90 inhibition affects shuttling protein delivery of ubiquitinated proteins to the proteasome by immunofluorescence assay.
9. Please design corresponding experiments to confirm the conclusion mentioned in the paper "Excessive chromatin PARylation due to PARG inhibition might affect optimal BRD4–histone binding, enabling PROTACs to out-compete the histone-binding pocket of BRD4."

Response to the reviewers' comments

We appreciate the reviewers' and editor's constructive suggestions, which helped us strengthen our study. In the revised manuscript, we have carefully addressed the reviewers' comments by providing substantial new experimental evidence, including 20 new figure panels, as described in this point-by-point response letter. We have also added discussions or revised the text according to the reviewers' suggestions. We thank the reviewers, as well as the editor, for their insightful comments.

Major improvements are as follows:

1) Generality of enhancement of targeted degradation other than BRD family proteins

Reviewers 1 and 2 asked whether the presented concept is applicable to neosubstrates other than BRD family proteins. We found that targeted degradation of another chromatin regulator, CDK9, is enhanced by PDD00017273 (hereafter, PDD) as well as by GSK2606414 (hereafter, GSK), showing that the enhancement of targeted degradation is not limited to BRD family proteins. We also showed that HSP90 inhibition exhibited an additive effect in the degradation of c-MET, CDK9, and ER- α .

Regarding the generality, because we have established a framework for identifying signaling pathways regulating targeted protein degradation, we believe that our presented concept will lead to the identification of new regulatory pathways using other neosubstrates in general.

2) Mechanism of action of signal inhibitors

Reviewers 1 and 3 advised that BRD4 dissociation from chromatin induced by PARG inhibition should be validated by, e.g., ChIP-Seq analysis. We performed ChIP-Seq analysis and found that BRD4 binding regions are clustered into MZ1-responsive (~80%) and MZ1-hyporesponsive (~20%) regions. We found that PDD indeed promotes the dissociation of BRD4 from chromatin in MZ1-responsive regions.

Regarding the mechanism of PERK or HSP90 inhibition, we have added several new results supporting the model that PERK/HSP90 inhibition enhances BRD4 degradation pre/post-ubiquitylation step, respectively. We are starting a new project to elucidate the detailed mechanism of action of these pathways in targeted protein degradation. Because the present study focused on PARG inhibition to elucidate the molecular mechanism, we would prefer to report the detailed mechanism of PERK/HSP90 inhibition in a future study.

3) We have carefully addressed all the comments raised by the reviewers and have reorganized the text and figures according to the reviewers' suggestions: New figures include Figs. 3g, 3h,

3i, 3j, 3k, 3l, 3m, 4d, 5h, 6f, 6g, 6h, 6i, and 6j, and Extended Data Figs. S2a, S2d, S3a, S3f, S3g, and S3h.

Reviewer #1 (Remarks to the Author):

Mori et al. performed a chemical screen for enhancers or abrogators of BRD4 degradation using a Hibit Knock-In reporter cell line and the PROTAC MZ-1. They identified the PARG inhibitor PDD17273, luminespib and GSK2606414 as enhancers of BRD4 degradation and proteasome inhibitors as abrogators of BRD4 degradation and confirmed these with follow up studies. They demonstrate that PDD17273 is a BRD specific enhancer, via inhibition of PARG that functions via enhanced ubiquitination of BRD4, especially by TRIP12 induced K29/K48 branched ubiquitination, resulting from enhanced availability of BRD4. Additionally, GSK2606414 enhances degradation via PERK inhibition, resulting in enhanced ubiquitination and luminespib acts by HSP90 inhibition, post ubiquitination event. The authors go on to demonstrate that both PDD17273 and GSK2606414 co-treatment with MZ-1 results in enhanced apoptosis, reduced proliferation and enhanced anti-tumor effect in vivo. Finally, the authors demonstrate that these compounds also enhance the activity of more active BRD PROTAC SIM1.

This is an interesting and well controlled study into the effects of intrinsic cell signaling pathways and how they can be modulated to enhance the action of PROTACS. However, there are several issues which prevent it from being suitable for the general audience of Nature Communications in this form.

We are pleased to read the reviewer's overall positive evaluation of our study. We also thank the reviewer for their important suggestions, which helped us substantially improve our manuscript. We have addressed the reviewer's comments point-by-point, as described below.

Point-1

The impact of these discoveries is limited due to the fact they are limited to BRD family proteins. The authors must demonstrate that the additive effects of these compounds is general across other protein targets for TPD (Eg. Kinases) otherwise the interest is limited. As the authors say, PDD17273 is likely specific to BRD family proteins but PERK and HSP90 inhibitors are likely to be generally useful. Especially since it was reported several years ago, and is well known, that HSP90 inhibitors enhance PROTAC activity (PMID: 29129716) further reducing the novelty of this study.

Response-1:

We appreciate the reviewer's suggestion. In the revised manuscript, we have carefully assessed whether the signal inhibitors also promote certain other neosubstrates such as kinases.

The PARG inhibitor

For the PARG inhibitor PDD17273, given its mechanism of action through chromatin modification, the enhancing effect of PDD might be restricted to chromatin regulators. Consistently, PDD did not enhance the degradation of non-chromatin neosubstrates such as ER- α (previous Fig. 2g). Therefore, we focused on CDK9, a chromatin-localized kinase regulating RNAPII-mediated transcription. We found that PDD enhanced CDK9 degradation induced by a CRBN-based PROTAC ThalSNS (Fig. 6f). Moreover, PDD significantly enhanced the ThalSNS-induced cell death in the ThalSNS concentration range 30–500 nM (Fig. 6g). These results suggest that PARG inhibition is effective for not only the degradation of BRD family proteins but also the degradation of another chromatin regulator, CDK9.

The HSP90 inhibitor

As the reviewer suggested, the effect of HSP90 inhibitors is likely to be more general. The paper the reviewer noted (PMID 29129716; we have cited the paper) reported that HSP90 inhibition destabilizes the kinase c-MET and that the co-treatment of HSP90 inhibitor with the PROTAC shows an additive effect on c-MET degradation. Therefore, we analyzed the effect of HSP90 inhibitor luminespib on the PROTAC-induced degradation of c-MET, ER- α , and CDK9. As a result, we found that luminespib decreased the steady-state levels of c-MET, ER- α , and CDK9 (Figs. 6h–j), as previously reported for c-MET and ER- α . The co-treatment of luminespib and PROTACs showed an additive effect on their degradation (Figs. 6h–j).

Collectively, our data suggest that the effect of HSP90 inhibition on PROTAC-induced degradation is not restricted to BRD family proteins. As the reviewer pointed out, the additive effect of HSP90 inhibitors and PROTACs on the degradation of HSP90-client proteins might not be novel per se. However, HSP90 inhibition alone does not induce destabilization of BRD4 and BRD2 (Fig. 4d ; see also lanes 1 and 7 of Fig. 4b). We also found that HSP90 associates with BRD4 at the normal state (Fig. S3h). We presume that BRD4 is a "silent" client of HSP90, which associates with HSP90 but, because of its low turnover rate, does not require HSP90 for its stabilization at the normal state. When BRD4 is ubiquitylated by an exogenous E3, the destabilization of BRD4 appears to be counteracted by HSP90. Therefore, we envision that the identification of HSP90 inhibitors as enhancers of BRD4 degradation might broaden the targets of HSP90 inhibition to include such silent clients.

The PERK inhibitor

We also investigated whether PERK inhibition enhanced degradation of other neosubstrates. GSK treatment enhanced the degradation of CDK9 but not ER- α or c-MET (Fig. 6h-j). Although the mechanism by which GSK co-treatment enhances the degradation of chromatin regulators is currently unknown, our new results indicate that the effect of PERK inhibition is not restricted to BRD family proteins but can be applied to another chromatin regulator, CDK9.

[Editorial note: this information was redacted, as it contained unpublished data]

We note that the signaling pathways we have identified here are not the "pan" regulators of targeted protein degradation, which might imply that signaling pathways affecting targeted protein degradation are diverse, depending on each neosubstrate. However, because we have established a framework for identifying such signaling pathways, we believe that the present study will lead to identification of new pathways using other neosubstrates of interest in general.

In summary, we provide evidence that 1) PDD and GSK promote targeted degradation of not only BRD family proteins but also another chromatin regulator, CDK9, and 2) luminespib promotes targeted degradation of CDK9, c-MET, and ER- α . We show that our concept of enhancing targeted protein degradation by inhibiting intrinsic signaling pathways is not restricted to BRD4. Moreover, the framework we established here for identifying PROTAC enhancers will be readily applicable to other neosubstrates in general.

Point-2

Some of the conclusions drawn are not fully supported by data and should be further explored:

1. The authors state that PDD17273 increases ternary complex formation and measure this by Cul2 Co-IP – this is hard to conclude from the figure presented in Fig. 3G. The authors should explore this further, potentially by NanoBRET ternary complex formation assays (PMID: 34432243)

Response-2:

We appreciate the reviewer's suggestion. The complex formation of BRD4 and CUL2 judged by co-IP-WB is a good indication of ternary complex formation, as established by Alabi et al. (PMID 33568647). However, we noticed that our previous co-IP-WB data might not be quantitative. Therefore, we conducted a label-free quantification of BRD4 immunocomplex using LC-MS. We have previously established that mass spectrometric quantification is a highly reliable technique to quantify the complex formation *in cell* (Tsuchiya et al., PMID 28525741). From three independent experiments, we found that the association of CUL2 with BRD4 is significantly enhanced by the co-addition of PDD (Fig. 3h). The results of conventional co-IP-WB (three biological replicates) are also provided in Fig. 3g and Fig. S2a.

We also attempted to carry out NanoBRET ternary complex formation assays (PMID: 34432243). However, while we acknowledge that NanoBRET is effective for analyzing *in vitro* interactions, the application of NanoBRET for *in cell* interactions appears to need further technical refinement in our hands. Nonetheless, we believe that the mass spectrometric quantification we have conducted here is a highly reliable technique to quantify the complex formation *in cell*.

Point-3

2. Further, the conclusions that increased PARylation evicts BRD4 from chromatin is insufficiently supported, additional experimentation (imaging, CHIP-Seq) is required to support this claim.

Response-3:

We agree with the reviewer's view that our model in which excessive PARylation decreases chromatin association of BRD4 should be rigorously analyzed using another methodology. According to the reviewer's suggestion, we have conducted ChIP-Seq analyses. The cells were treated either with control or PDD, or with MZ1 as a positive control, and soluble chromatin were subjected to immunoprecipitation using anti-BRD4 antibody.

First, the BRD4 binding regions were divided into three clusters. In clusters 1 and 2, which account for 1.4% and 18% of the total binding regions, respectively, MZ1 did not induce chromatin dissociation of BRD4 (Fig. 3i–k). These regions are highly enriched with histone H3K27 acetylation (Fig. 3l). The data suggest that clusters 1 and 2 are "MZ1-hyporesponsive" regions, in which BRD4 appears to tightly associate with high levels of H3K27Ac. However, in cluster 3, which accounts for 80% of the total binding regions, MZ1 induced eviction of BRD4 from chromatin. These regions are "MZ1-responsive" regions.

Strikingly, in cluster 3 (MZ1-responsive regions), PDD massively decreased chromatin association of BRD4 (Fig. 3k). The data support our model in which excessive PARylation decreases chromatin association of BRD4. Because cluster 3 is ~80% of the total BRD4 binding regions (Fig. 3j), we assume that the PDD-induced chromatin eviction of BRD4 in cluster 3 is the main contributor to the enhanced BRD4 ubiquitylation and degradation.

We also found that, in clusters 1 and 2 (MZ1-hyporesponsive regions), PDD increased the chromatin association of BRD4. Although the reason is currently unknown, excessive PARylation can alter chromatin structures to enhance BRD4 accessibility. We presume that these BRD4 associating with the cluster 1/2 regions do not contribute to the MZ1-induced degradation, at least within a short timecourse. Notably, although beyond the scope of the current study, these observations introduce a possibility that direct manipulation of histone acetylation might facilitate complete eviction and subsequent degradation of BRD4 tightly associated with chromatin; this hypothesis warrants further examination in our future project.

Point-4

No protocol or method is presented for how the screen was performed and which compounds were screened – this must be included in the manuscript or SI for it to be useful to the scientific community. Compounds which failed to induce changes in BRD4 degradation could be as informative in understanding mechanisms, as those that did (depending on library composition).

Response-4:

We have now included the detailed method for screening in the supplementary methods section. We also included the list of compounds in the supplementary information (Table S1).

Point-5

4 points are shown on Fig. 1E for compounds which inhibited PROTAC activity but only 3 are named and all are proteasome inhibitors, what is the fourth compound? Depending on the compound library, several other compounds should be included in this group (eg. JQ1, MLN4924 etc.)

Response-5:

The fourth compound is "Ixazomib (MLN2238)," another proteasome inhibitor. The name has been added to Fig. 1e. Unfortunately, our library does not include JQ1 and MLN4924. Because the present study is a proof-of-concept work to identify regulatory signaling pathways, we are currently starting to explore a larger drug library, including FDA-approved drugs.

Point-6

Statistics should be included in Fig. 5H to report the significance of this relatively subtle difference.

Response-6:

[Editorial note: this information was redacted, as it contained unpublished data]

Reviewer #2 (Remarks to the Author):

Key results

The key message of this study centers around the enhancement of BRD4 (and BRD2/3) degradation by VHL and CRBN PROTACs through targeted inhibition of the PAR and unfolded protein response pathways (PERK and HSPs). The authors performed a screen enabled by a stable HiBiT-Tag BRD4 cell line of various inhibitors. This is either a direct effect on BRDs through induction of their chromatin dissociation or by influencing degradation post-ubiquitination. **These types of cell intrinsic effects may broadly apply to targeted degradation of specific substrates in general and allow for better mechanistic understanding in the field.**

Validity

The story focused on the degradation of BRD4 is done appropriately, with the proper UPS controls and would be easily reproducible by readers of the study. The methodology they present can be applied more broadly to other neo-substrates. The data presented in the paper is of high quality and easy to follow, with significant impacts on protein degradation clear from the author's figures. How generalizable these conclusions are outside of Bromodomain proteins is unclear. While the steady state of the BRD proteins stays the same, changes to their intrinsic half-life by the GSK and PDD molecules may also have impacts on their degradation.

Significance

The concepts presented in this paper will have a significant impact on the scientific community as it seeks to elucidate the impacts of variable cell signaling events on targeted protein degradation. This touches on a critical biological aspect that influences the activity of PROTAC molecules distinct from previous studies that have looked at E3 ligase expression, target expression, and localization. **Though the scope of the paper is narrowly focused on BRD4, the techniques can be easily applied to other targets. Understanding these pathways may impact our knowledge of the development of resistance to degraders and their treatment.**

We are pleased to read the reviewer's positive evaluation of our study, especially his/her view that "These types of cell intrinsic effects may broadly apply to targeted degradation of specific substrates in general and allow for better mechanistic understanding in the field" and "Though the scope of the paper is narrowly focused on BRD4, the techniques can be easily applied to other targets. Understanding these pathways may impact our knowledge of the development of resistance to degraders and their treatment."

We also thank the reviewer for their important suggestions, which helped us substantially improve our manuscript.

To address the comment "How generalizable these conclusions are outside of Bromodomain proteins is unclear." we have extensively analyzed the degradation of other targets such as CDK9, c-MET, and ER- α . We provide evidence that the effects of PDD/GSK/luminespib on targeted protein degradation is not restricted to BRD family proteins. For details, please see **Response-1 to Reviewer-1**.

For the comment "While the steady state of the BRD proteins stays the same, changes to their intrinsic half-life by the GSK and PDD molecules may also have impacts on their degradation." please refer to Response-2 below.

We have addressed the reviewer's comments point-by-point, as described below.

Data and methodology

Analytical approach

Your assessment of the strength of the analytical approach, including the validity and comprehensiveness of any statistical tests. If any aspect of the analytical approach is outside the scope of your expertise, please note this in your report or in the comments to the editor.

Suggested improvements

Point-1

I think the authors should consider changing the title of this paper due the bromodomain focus of the conclusions. If it included analysis of other PROTAC molecules from literature this would be more appropriate.

Response-1

In the revised manuscript, we have added the analysis of other PROTACs and their targets and have found that the signal inhibitors we analyzed here facilitate targeted degradation of other targets such as CDK9, c-MET, or ER- α (please see **Response-1 to Reviewer-1**). We would like to follow the reviewer's suggestion if it appears better to change the title.

Point-2

The authors demonstrate that the inhibitors do not affect the steady state levels of the

bromodomain proteins, another consideration would be to investigate how the inhibitors influence the basal half-life of BRD2/3/4 with a cycloheximide chase or similar assay.

Response-2

We appreciate the reviewer's insightful suggestion. In the revised manuscript, we have performed cycloheximide chasing. We found that BRD4 as well as BRD2/3 are highly stable in the steady state. Even after treatment with CHX for 14 h, which is the longest CHX incubation duration before the cells exhibit clear toxicity in our hands, we did not observe a clear decrease in the protein levels of BRD2/3/4 (Fig. 4d). After CHX incubation for 6 and 14 h, treatment of cells with PDD, GSK, or luminespib did not change the half-life of BRD4/2. We conclude that, within the technically available time ranges, the signal inhibitors we investigated do not influence the basal half-life of BRD2/3/4.

Point-3

The authors show the synergy observed with co-treatment of GSK/PDD with MZ1 (Figs. 5b and S2). It would also be important to show this with JQ1 and GSK/PDD co-treatment as a control, as this should not result in the same synergistic effect if the synergy is due to degradation impacts vs downstream signaling. These controls would also be nice to see in the impact on downstream signaling in figure 5g.

Response-3

Taking the reviewer's advice, we analyzed whether the co-treatment of PDD/GSK causes synergistic effects with JQ1. Strikingly, JQ1-induced cell death was not enhanced by the co-treatment of PDD/GSK (Fig. S3f-g). In addition, neither the JQ1-induced increase of p21 and OSGIN expression, nor the decrease of ERBB2 and MYC, was further enhanced by the co-treatment of PDD/GSK (Fig. 5h). Such gene expression patterns of JQ1 co-treatment clearly differ from those of MZ1 co-treatment (Fig. 5g). Collectively, these results underscore the validity of our proposed mechanism in which these signal inhibitors accelerated the targeted degradation of BRD family proteins.

Point-4

More detail of the diversity of the screening library may be useful.

Response-4

We have provided details of the screening library in Table S1.

Point-5

The mechanism of the enhancement of the VHL-BRD ternary complex in Fig. 4h is unclear, do you see the same linkage changes?

Response-5

We thank the reviewer for his/her insightful comment. In the case of the Ub linkage types, our previous study established that the majority of ubiquitin linkages conjugated on BRD4 upon MZ1 treatment are K29- and K48-linked chains (Kaiho-Soma et al., PMID 33567268). Therefore, we have analyzed the modification of BRD4 with these two linkage types. K29- and K48-linked ubiquitylation of BRD4 and BRD2 was enhanced by the co-treatment of GSK but not by that of luminespib (Fig. 4g–h). Therefore, these results suggest that certain GSK-induced change(s) in the cellular environment favor the formation of the BRD4-PROTAC-CRL2 ternary complex.

In the case of new data regarding the mechanism of enhanced BRD4 degradation by GSK/luminespib, we are starting a new project to reveal the detailed mechanism of action (please refer to Point-1 of Reviewer-1). However, as Reviewer-2 kindly commented, we also believe that the concept of controlling the targeted protein degradation by manipulating cell intrinsic pathways might "broadly apply to the targeted degradation of specific substrates in general and allow for better mechanistic understanding in the field." We would prefer to analyze the detailed molecular mechanism for GSK/luminespib in BRD4 degradation in our future project.

Point-6

Correlating the enhancement of degradation by GSK or PDD with the expression of their respective targets may give insight to how the variance of these factors impact degradation.

Response-6

The results of upregulated/downregulated genes induced by GSK or PDD analyzed by RNA-Seq are now presented in the supplementary information (Table S3). From a gene ontology analysis, we did not see specific enrichment of any pathways. Also, we did not observe the up/down-regulation of Ub ligases or DUBs.

As described in Response-1 to Reviewer-1, we conducted TMT-based quantitative proteomics and found certain interesting candidates that might mediate the effects of GSK on accelerated BRD4 degradation. We would like to analyze these mechanisms in detail in our future project.

Point-7

Although outside of the scope of this work, analysis of genomic CRISPR screening data may provide further insights as to the most important pathways that enhance targeted protein degradation as this will cover more than the inhibitor library.

Response-7

We appreciate the insightful comment by the reviewer. We also wanted to conduct a CRISPR screening to identify pathways regulating targeted protein degradation. We are currently preparing for such a screening as our next project.

Clarity and context

The text of this paper is presented clearly.

References

This manuscript references previous literature appropriately.

Reviewer #3 (Remarks to the Author):

In this study, the authors identify cellular signaling pathways that modulate the targeted degradation of the anticancer target BRD4 and related hard-to-degrade targets BRD2/3 induced by CRL2VHL- or CRL4CRBN -based PROTACs. Several inhibitors of cellular signaling pathways were screened as degradation enhancers, such as poly-ADP ribosylation (PARG inhibitor PDD00017273), unfolded protein response (PERK inhibitor GSK2606414), and protein stabilization (HSP90 inhibitor luminespib). Among them, PARG inhibition promotes TRIP12-mediated K29/K48-linked branched ubiquitylation of BRD4 by facilitating chromatin dissociation of BRD4 and formation of the BRD4-PROTAC-CRL2VHL ternary complex; HSP90 inhibition promotes BRD4 degradation after the ubiquitylation step. In addition, these signal inhibitors sensitize cells to PROTAC-induced apoptosis. However, some issues should be addressed:

We thank the reviewer for their important suggestions, which helped us substantially improve our manuscript. We have addressed the reviewer's comments point-by-point, as described below.

Point-1

1. The proteasome inhibitor in Figure S1b is Carfilzomib, while the proteasome inhibitor mentioned in the paper is MG132.

Response-1

We apologize for the mistake. 'Carfilzomib' is correct for the data. We have now corrected the text.

Point-2

2. No statistical analyses were made in Figure 2a and Figure 2b

Response-2

For Fig. 2a, we conducted a statistical analysis (one-way ANOVA) from three independent assays, which revealed that the PDD treatment significantly enhanced BRD4 degradation at 1, 2, and 4 h timepoints.

For Fig. 2b, we have included the results of one-way ANOVA. The degradation of HiBiT-BRD4 is significantly enhanced by the co-treatment of $\geq 0.3 \mu\text{M}$ PDD, whereas cell viability is significantly decreased by $\geq 10 \mu\text{M}$ PDD.

Point-3

3. In Figure 3a, the concentration of MZ1 changes, while the time of PDD changes. Multiple variables lead to unreasonable experimental design.

Response-3

We apologize for the confusing labeling. The concentration of MZ1 changes, whereas the concentration and duration of PDD are the same (3 μ M, 6 h). To avoid reader misunderstanding, we have changed the labeling.

Point-4

4. There was no statistical analysis of the changes of BRD4–MZ1–CRL2VHL ternary complex assembly in Figure 3g.

Response-4

We have conducted a label-free quantification of the FLAG-BRD4 immunocomplex using LC-MS, and the quantities of co-purified CUL2 from three independent experiments were analyzed using ANOVA. As shown in Fig. 3h, the MZ1-induced interaction of CUL2 is significantly enhanced ($P < 0.0001$) by the MZ1+PDD co-treatment.

Point-5

5. Please replace Figure 4h with a better-quality version where all values can be read clearly.

Response-5

We have replaced Fig. 4h with a better-quality version (now Fig. 4i).

Point-6

6. Excessive tumor volume at the end of treatment in a mouse allograft model.

Response-6

[Editorial note: this information was redacted, as it contained unpublished data]

Point-7

7. The starting point of HCT116 cell viability assay in Figure 5b (ii) was significantly different when MZ1 concentration was 0 nM.

Response-7

As the reviewer pointed out, GSK treatment alone (the point of 0 nM MZ1) slightly decreased cell viability. To confirm whether GSK co-treatment sensitizes cells to MZ1, we analyzed the percentages by which the cell number decreased as a result of treatment with MZ1. As shown in Fig. S3a, GSK co-treatment significantly sensitizes cells to MZ1 at MZ1 concentrations from 30 to 300 nM.

Point-8

8. Please verify whether HSP90 inhibition affects shuttling protein delivery of ubiquitinated proteins to the proteasome by immunofluorescence assay.

Response-8

To investigate whether HSP90 inhibition affects shuttling protein delivery of ubiquitylated proteins to the proteasome, we focused on RAD23B because RAD23B is the major shuttling protein associating with the proteasome and mediating proteasomal degradation (Yasuda et al., PMID 32025036). We first conducted an immunofluorescence assay; however, because RAD23B, HSP90, the proteasome, and ubiquitylated proteins are all stained diffusely, we could not verify the co-localization. Therefore, we conducted a co-immunoprecipitation analysis using anti-FLAG antibody from the 293T cells expressing FLAG-RAD23B. Neither the interaction of RAD23B with the proteasome nor that of RAD23B with the ubiquitylated proteins was changed by HSP90 inhibition (Fig. S2d). These results suggest that HSP90 inhibition does not globally affect shuttling protein (RAD23B) delivery of ubiquitylated proteins to the proteasome.

Regarding the mechanism of accelerated BRD4 degradation by HSP90 inhibition, we found that HSP90 interacts with BRD4 at the steady state (Fig. S3h). Please see Response-1 to Reviewer-1.

Point-9

9. Please design corresponding experiments to confirm the conclusion mentioned in the paper

“Excessive chromatin PARylation due to PARG inhibition might affect optimal BRD4–histone binding, enabling PROTACs to out-compete the histone-binding pocket of BRD4.”

Response-9

We appreciate the reviewer's insightful comment. We agree with the reviewer that whether PARG inhibition affects chromatin association of BRD4 should be rigorously examined. Therefore, we conducted a ChIP-Seq analysis. The cells were treated either with control or PDD, or with MZ1 as a positive control, and soluble chromatin was subjected to immunoprecipitation using anti-BRD4 antibody.

First, the BRD4 binding regions were divided into three clusters. In clusters 1 and 2, which account for 1.4% and 18% of the total binding regions, respectively, MZ1 did not cause chromatin dissociation of BRD4 (Fig. 3i–k). These regions are highly enriched with histone H3K27 acetylation (Fig. 3l). The data suggest that clusters 1 and 2 are "MZ1-hyporesponsive" regions, in which BRD4 appears to tightly associate with high levels of H3K27Ac. However, in cluster 3, which accounts for 80% of the total binding regions, MZ1 induced eviction of BRD4 from chromatin. These regions are "MZ1-responsive" regions.

Strikingly, in cluster 3 (MZ1-responsive regions), PDD massively decreased chromatin association of BRD4 (Fig. 3k). The data support our model that excessive PARylation decreases chromatin association of BRD4. Because cluster 3 is ~80% of the total BRD4 binding regions (Fig. 3j), we assume that the PDD-induced chromatin eviction of BRD4 in cluster 3 mainly contributes to the enhanced BRD4 ubiquitylation and degradation.

We also found that, in clusters 1 and 2 (MZ1-hyporesponsive regions), PDD rather increased chromatin association of BRD4. Although the reason is currently unknown, excessive PARylation might alter chromatin structures to enable BRD4 accessibility. We presume that these BRD4 associating with the cluster 1/2 regions do not contribute to the MZ1-induced degradation, at least within a short timecourse. We note that, although outside the scope of the current study, these observations introduce the possibility that direct manipulation of histone acetylation might facilitate complete eviction and subsequent degradation of BRD4 tightly associated with chromatin; this hypothesis warrants examination in our future project.

In addition, we analyzed the chromatin fraction using anti-PAR immunoblotting and showed that PARG inhibition induces excessive PARylation of chromatin proteins (Fig. 3m).

Collectively, our new data support our model that excessive chromatin PARylation induces dissociation of BRD4 from chromatin, leading to enhanced BRD4-PROTAC-CRL2 ternary complex formation and subsequent BRD4 ubiquitylation. The dissociation of BRD4 bromodomain from acetylated histone likely favors the binding of the PROTAC to BRD4. However, as the reviewer pointed out, strict proof of these causal relationships *in cell* is technically challenging at this stage. Therefore, we have toned down our claims related to this issue as follows:

"Using ChIP-Seq analysis, we showed that excessive chromatin PARylation due to PARG inhibition indeed accelerates BRD4 dissociation from chromatin. This accelerated dissociation likely facilitates binding of PROTACs to BRD4, leading to enhanced ternary complex formation."

Reviewers' Comments:

Reviewer #1:

Remarks to the Author:

The authors have adequately addressed my concerns and significantly improved their manuscript.

Reviewer #2:

Remarks to the Author:

The extensive modifications made to the manuscript by the authors meets the criteria for publication in Nature Communications. The authors revisions and rebuttal provide satisfactory responses to the majority of points raised by our review as well as those of Reviewers 1 and 3. Their work to expand the concept outside of BRD proteins to CDK9, ER-alpha and c-MET by the addition of new data and discussion addressed a major concern of the scope of the work outside of bromodomain proteins. The title of the article now accurately fits the conclusions. The authors added additional controls to demonstrate the activity observed was not due to intrinsic changes to protein stability upon inhibition of PARG or HSP90. They also addressed mechanistic questions relating to inhibition vs. degradation with their investigation of JQ1. The new ChIP-Seq analysis delves further into the activity of the PARG inhibition and its impact on global chromatin PARylation which will affect the PROTAC mediated degradation of many other proteins. These new findings add to the significance of the publication and additional suggestions on the GSK ligand would be outside of the scope of this publication.

Signed: Ryan Potts, Amgen

Reviewer #3:

Remarks to the Author:

This revised manuscript has addressed all issues I raised and should be accepted for publication in Nature Communications.